# Drivers and implications of alternative routes to fuels decarbonization in net-zero energy systems

Bryan K. Mignone [1] ✉, Leon Clarke[2,3], James A. Edmonds [4], Angelo Gurgel [5], Howard J. Herzog [5], Jeremiah X. Johnson [6], Dharik S. Mallapragada [5], Haewon McJeon[4], Jennifer Morris [5], Patrick R. O'Rourke [3,4], Sergey Paltsev [5], Steven K. Rose[7], Daniel C. Steinberg[8] & Aranya Venkatesh[9,10]

Energy transition scenarios are characterized by increasing electrification and improving efficiency of energy end uses, rapid decarbonization of the electric power sector, and deployment of carbon dioxide removal (CDR) technologies to offset remaining emissions. Although hydrocarbon fuels typically decline in such scenarios, significant volumes remain in many scenarios even at the time of net-zero emissions. While scenarios rely on different approaches for decarbonizing remaining fuels, the underlying drivers for these differences are unclear. Here we develop several illustrative net-zero systems in a simple structural energy model and show that, for a given set of final energy demands, assumptions about the use of biomass and $CO_2$ sequestration drive key differences in how emissions from remaining fuels are mitigated. Limiting one resource may increase reliance on another, implying that decisions about using or restricting resources in pursuit of net-zero objectives could have significant tradeoffs that will need to be evaluated and managed.

Scenarios consistent with global climate stabilization are characterized by systems with declining greenhouse gas (GHG) emissions, with net-zero carbon dioxide ($CO_2$) emissions typically attained around mid-century for temperature outcomes near 1.5 °C and in the 2070s for temperature outcomes near 2 °C[1]. The role of end use electrification and deep decarbonization of electricity generation has received extensive consideration in the literature[1–9]. At the same time, significant amounts of liquid fuels and natural gas remain in most scenarios at the time of net-zero emissions[10]. This result occurs because it may be difficult or costly to electrify some sectors, so retaining some fuels while mitigating their emissions may be more cost-effective than other options[11]. However, the specific ways in which these remaining

fuels attain net-zero emissions—and the drivers underlying different pathways—has received less consideration. This gap could limit the applicability of such scenarios to policy, planning, and investment decision-making related to fuels decarbonization, which is a key component of economy-wide mitigation.

Once end use substitution (e.g., electrification) has occurred, there are several approaches to mitigate emissions from remaining fuels—specifically from liquid hydrocarbon fuels and natural gas. One approach is to produce alternative fuels using biomass as a feedstock (biofuels). Another approach is to combine $CO_2$ that is removed from the atmosphere with hydrogen to make a synthetic hydrocarbon fuel not derived from biomass (synthetic fuels). A third option is to

[1]ExxonMobil Technology and Engineering Company, Annandale, NJ, USA. [2]Bezos Earth Fund, Washington, DC, USA. [3]School of Public Policy, University of Maryland, College Park, MD, USA. [4]Pacific Northwest National Laboratory, Joint Global Change Research Institute, College Park, MD, USA. [5]Massachusetts Institute of Technology, Cambridge, MA, USA. [6]Department of Civil, Construction, and Environmental Engineering, North Carolina State University, Raleigh, NC, USA. [7]Energy Systems and Climate Analysis, EPRI, Washington, USA. [8]National Renewable Energy Laboratory, Golden, CO, USA. [9]Department of Engineering and Public Policy, Carnegie Mellon University, Pittsburgh, PA, USA. [10]Present address: Energy Systems and Climate Analysis, EPRI, Washington, DC, USA. ✉e-mail: bryan.k.mignone@exxonmobil.com

continue using conventional, fossil-derived fuels while offsetting the associated emissions using carbon dioxide removal (CDR) technologies such as bioenergy with carbon capture and sequestration (BECCS) or direct air capture (DAC) with $CO_2$ sequestration (DACCS). While each of these approaches can be found within existing scenarios, there is no clear consensus about the volume of fuels that would remain in net-zero energy systems or the mix of approaches that would be utilized to mitigate emissions from remaining fuels. The goal of this paper is to fill this gap by focusing on the drivers and implications of alternative routes to emissions mitigation from remaining fuels in net-zero energy systems.

A variety of modeling frameworks have been used to produce energy transition scenarios consistent with net-zero emissions objectives, including Integrated Assessment Models (IAMs)[12,13] and Macro Energy System (MES) models[14]. IAMs exhibit broader geographic and sectoral scope, linking climate, land, energy and economic systems globally, and have been used extensively to produce energy transition scenarios used in global assessments[1,15] as well as scenarios of regional and national GHG mitigation[16,17]. MES models tend to have greater spatial and temporal resolution and have been used widely to produce scenarios of net-zero energy systems for several different countries and regions[6,18–24].

Regardless of the framework used to produce them, several features are common to most net-zero scenarios, such as demand-side changes, including greater deployment of energy efficiency and sufficiency measures, increased electrification of end uses, near elimination of emissions from electricity production, reduction in the volumes of liquid fuels (liquids) and natural gas consumed, even steeper declines in coal use, and deployment of CDR technologies, often in conjunction with expanded use of biomass, to offset remaining emissions[25–27]. Examining the median values in the set of 97 (C1) scenarios assessed by the Intergovernmental Panel on Climate Change (IPCC) that are consistent with 1.5 °C stabilization with limited or no overshoot[1,28], the amount of electricity in final energy (globally) is projected to increase from 84 EJ in 2020 to 207 EJ in 2050 (interquartile range of 178–252 EJ), and the share of electricity from wind, solar, hydro, nuclear, biomass, and fossil with carbon capture and storage (CCS) is projected to increase from 36% to 95% (94–97%) over the same period. The median share of electricity in final energy in 2050 in the IPCC 1.5 °C scenarios is 52% (48–58%), compared to 20% in 2020. In the same scenarios, liquid fuels in final energy are projected to decrease from 161 EJ in 2020 to 91 EJ (60–122 EJ) in 2050, natural gas in final energy is projected to decrease from 65 EJ to 30 EJ (19–48 EJ), and coal in final energy is projected to decrease from 40 EJ to 2 EJ (1 to 7 EJ). At the same time, biomass primary energy is projected to increase from 31 EJ in 2020 to 105 EJ (84–143 EJ) in 2050, and engineered CDR (BECCS and DACCS) is projected to grow to about 4 Gt $CO_2$ (1.9–5.7 Gt $CO_2$) per year.

Although net-zero scenarios share many qualitative features, they differ in some respects, such as the extent to which liquid fuels and natural gas remain in end uses and how emissions from such fuels are managed or abated. Differences in the amounts of remaining liquid fuels and natural gas are apparent in the significant variation in those fuels in the 1.5 °C scenarios discussed above. To some extent these changes can be attributed to differences in the extent of electrification and differences in overall final energy demand (see Supplementary Table 2). Differences in mitigation approaches for remaining fuels are suggested by the variation in the use of biomass primary energy and CDR, given the relevance of these resources to mitigating emissions from fuels. The variation in biomass primary energy and CDR, in turn, reflects the uncertainty in how society may decide to use or limit these resources given other factors such as land and water use[29] as well as other institutional factors affecting their deployment[30].

While prior studies have examined differences in electrification[25] and remaining fuel use[10] at net-zero, few have explicitly examined the range of pathways available to mitigate emissions from remaining fuels even though significant volumes remain in most scenarios, including in those with higher levels of electrification. This study aims to fill this gap by showing that, for a given set of final energy demands, assumptions about the use of biomass and $CO_2$ sequestration drive key differences in how emissions from remaining fuels are mitigated. Because limiting one resource may increase reliance on another, decisions about using or restricting resources in pursuit of net-zero objectives could have significant tradeoffs that will need to be evaluated and managed. In addition to highlighting these tradeoffs for decision-makers, we also provide several actionable recommendations for the research community.

## Results

### Fuels-related emissions mitigation in existing studies

In net-zero systems, any emissions that would have resulted from remaining conventional liquid fuels or natural gas must be either directly abated (by switching to lower-carbon alternatives) or managed (by offsetting emissions with removals). Therefore, in a given scenario, total mitigation of emissions from remaining fuels—the amount of mitigation needed if all remaining liquid fuels and natural gas were produced from conventional, fossil-based sources—can be approximately decomposed into contributions from CDR (BECCS and DACCS), biofuels and biogas, and non-biomass synthetic fuels (which we will refer to simply as synthetic fuels). When biofuels or biogas are coupled with CCS, then the removal component is considered CDR. Figure 1 shows an approximate decomposition of total mitigation into these contributions for the global 1.5 °C scenarios in the IPCC AR6[1,28], the IEA global net-zero scenario[31], the set of U.S. scenarios developed as part of the Open Energy Outlook (OEO)[32–34], the set of U.S. scenarios developed for the Electric Power Research Institute (EPRI) Low Carbon Resource Initiative[35], and the set of U.S. scenarios examined by Williams et al.[6]. The contributions are expressed as a share of total estimated emissions mitigation from fuels in each scenario, so that studies with varying geographic scope can be compared.

Figure 1 reveals some notable similarities and differences between the scenarios within and across studies. In general, the IPCC 1.5 °C scenarios rely heavily on biomass-based options—BECCS or biofuels, depending on the amount of CDR. To some extent, this reflects the state of integrated assessment model development at the time of IPCC AR6; DAC and synthetic fuels have become a more recent focus. In contrast, two of the three OEO scenarios rely heavily on DACCS, as more CDR is deployed to effectively attain a net-zero GHG energy system while biomass supply approaches assumed limits. Williams et al.[6] relies more heavily on alternative fuels, with synthetic fuels providing a larger share of mitigation than biofuels in two of the three scenarios shown. The relatively higher use of BECCS in the Low Land case appears to reflect land constraints applied to multiple sectors (not only bioenergy production) and the relatively high removal efficiency (relatively low land use per ton removed) of some BECCS technologies.

However, in most of the scenarios shown in Fig. 1, except when strongly constrained, synthetic fuels do not deploy or deploy only modestly. For example, in the EPRI LCRI Limited Options scenario, synthetic fuels deploy only when geologic $CO_2$ storage is assumed not to be available and biomass supply is assumed to be limited, and in this case, the overall amount of remaining fuels is lower than in the other cases. In contrast, biofuels contribute to mitigation in all the cases shown, albeit to varying extents. Although the cases shown here represent only a subset of the scenarios in the literature that attain net-zero emissions by 2050, these scenarios are sufficient to illustrate key differences in potential mitigation pathways for remaining fuels. Scenarios differ not only in their relative reliance on CDR and alternative fuels, but also in whether those options utilize biomass-based or other synthetic routes. What is currently lacking is a unified explanation for these differences in terms of underlying drivers.

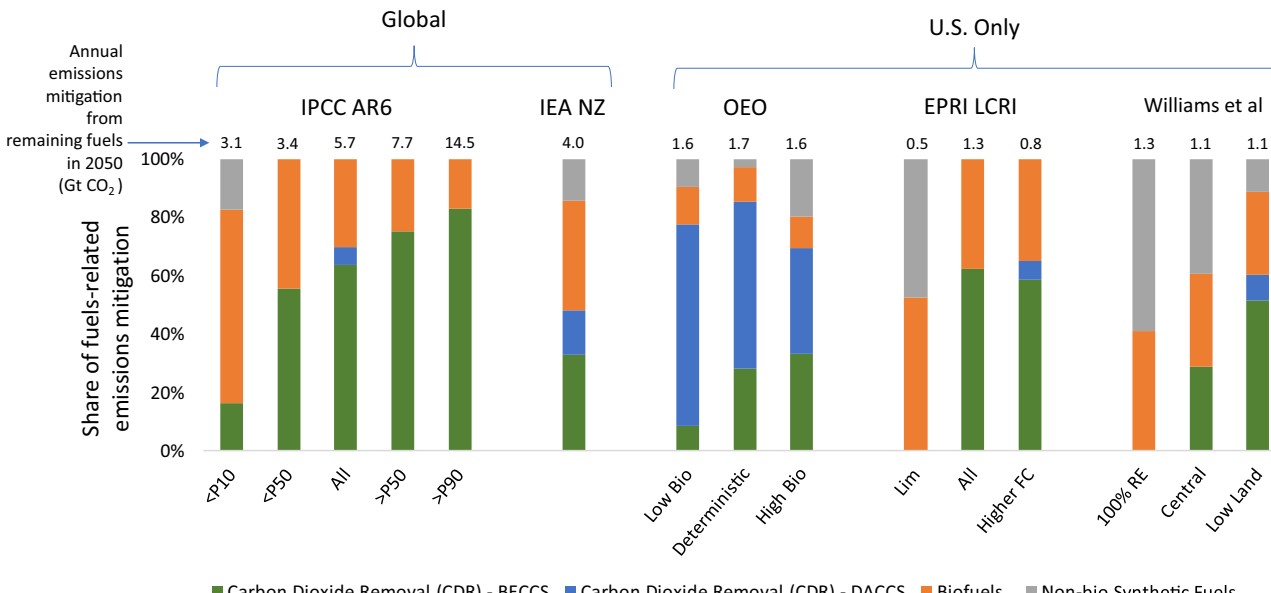

**Fig. 1 | Mitigation decomposition for remaining liquid fuels and natural gas in 2050 from five different sources.** Fuels-related mitigation refers to the amount of mitigation needed if all remaining liquid fuels and natural gas were produced from conventional, fossil-based sources. The Intergovernmental Panel on Climate Change (IPCC) bars summarize the 97 1.5 °C (C1) scenarios in the IPCC AR6 scenarios database[1,28]. The center bar uses median values from the full set of 1.5 °C scenarios, whereas the other bars use median values from subsets of scenarios in which Carbon Dioxide Removal (CDR) is less than or greater than a given percentile (reflected in the label). P10 is 10th percentile, P50 is 50th percentile, and P90 is 90th percentile. The International Energy Agency (IEA) NZ is the single net-zero scenario from IEA[31]. For the Open Energy Outlook (OEO), the left and right cases are the cases with 10th percentile and 90th percentile outcomes for BECCS deployment, respectively, and the central case is the single deterministic net-zero case from the OEO[34]. For the Electric Power Research Institute (EPRI) Low-Carbon Resources Initiative (LCRI), the left case is the Limited Options case, the middle case is the All Options case, and the right case is the Higher Fuel Cost case from Blanford et al.[35]. Williams et al. includes three scenarios from their U.S. net-zero study[6], with the center bar based on outputs from the central case, the left bar based on the case with the least amount of CDR (100% RE), and the right bar based on the case with the greatest amount of CDR (Low Land). The derived values using reported model outputs for all studies are shown in Supplementary Table 1. Note that the share of liquid fuels and natural gas in final energy in 2050 differs across the studies and scenarios shown. The shares range from 25–33% in the IPCC, 36% in IEA NZ, 29–31% in the OEO for the cases shown, 20–48% in EPRI LCRI, and 39–43% in Williams et al. for the scenarios shown. In 2020, the global share was 55%. *BECCS* Bioenergy with Carbon Capture and Sequestration. *DACCS* Direct Air Carbon Capture and Sequestration. Source data are provided as a Source Data file.

Differences in how emissions from remaining fuels are mitigated, whether in the same study or across studies, could be driven by several factors. One factor is the assumed availability of enabling technologies. For example, in the IPCC scenarios, synthetic fuels deployment may be lower largely because those technologies are not included in many models. Another factor is how available technologies—particularly early-stage technologies such as DAC or other enabling technologies—are parameterized[36,37]. A third factor is how constraints on resources, which may be related to physical limits but more generally reflect societal preferences, are specified. For example, biomass supply and CCS availability can significantly affect mitigation outcomes, including the cost of mitigation and the feasibility of meeting certain climate objectives[38,39]. In addition, regional differences in resources and preferences may explain differences in outcomes across regions[26].

Whether end use decisions are explicitly represented is another factor that would primarily affect the overall demand for fuels. Demand for fuels could also depend on additional assumptions, such as fossil fuel resources and prices, the availability, efficiency, and cost of substitutes (e.g., electricity and hydrogen consuming technologies in end uses), how end use decisions are represented, particularly at the consumer level, and the full range of policies implemented. Supplementary Table 1 reports remaining liquids and natural gas fuel volumes for the scenarios examined in Fig. 1, which vary considerably across studies.

As demonstrated by the range of results shown in Fig. 1, there is no easily identifiable consensus across the IAM and MES modeling communities about how emissions from remaining liquid fuels and natural gas would be mitigated in net-zero systems. BECCS, DACCS, biofuels,

and other synthetic fuels would likely require different enabling conditions and could have different implications for the broader energy system or other societal outcomes. In the following sections, we explore key drivers and implications of these alternative mitigation options using a simple global energy system model designed to shed light on the range of outcomes observed in scenarios produced by more complex models.

### Illustrative net-zero energy systems

To demonstrate how different mitigation options for remaining liquid fuels and natural gas may occur under alternative assumptions, we specify a simple (single region, single period) model of the global energy system. While not intended as a substitute for more complex models, this approach is highly transparent and allows us to clearly identify and evaluate relationships among key variables in a net-zero energy system that might not otherwise be apparent. Supply of electricity, hydrogen, liquids, and natural gas is determined via least-cost optimization to satisfy exogenous final demands for these energy carriers. Final energy demands are assumed to be the median 2050 values from the 1.5 °C (C1) scenarios in the IPCC AR6 database[1,28] (see Methods). A limitation of this approach is that end use substitution is not endogenously modeled. Rather, we effectively adopt an amount of end use substitution (e.g., conventional fuels switching to electricity or hydrogen) that reflects the extent of this substitution realized in the IPCC 1.5 °C scenarios. Although prices of electricity, hydrogen, liquid fuels, and natural gas may change across our scenarios, we do not adjust demand in response to changes in prices. Production technologies in each sector compete based on levelized cost, with cost and

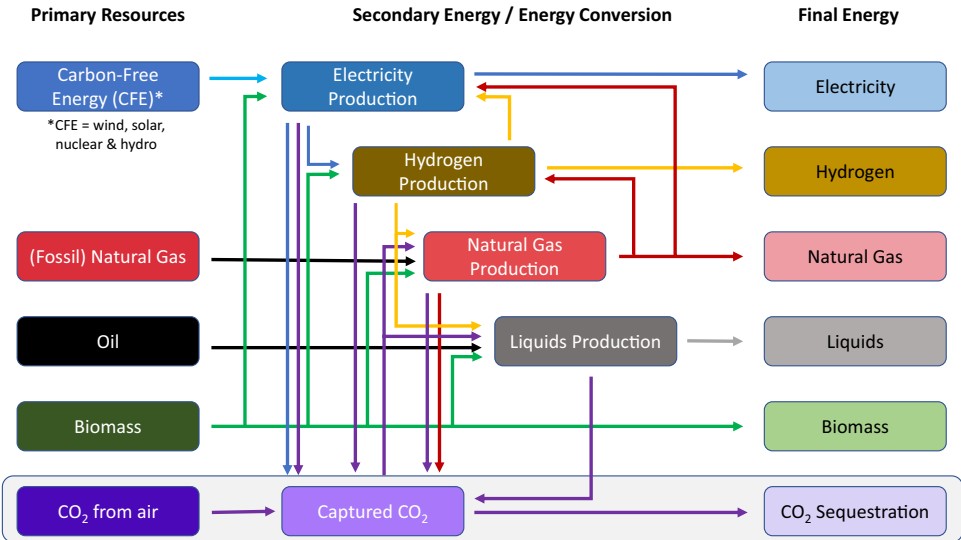

**Fig. 2 | Key components and linkages in the simple energy system model.** Primary resources (primary energy) are shown on the left, energy conversion (secondary energy) is shown in the center, and final energy is shown on the right. Supply and demand for captured $CO_2$ is shown in the bottom row. Lines with arrows represent inputs and outputs, with lines of the same color generally representing the same vector. Electricity is shown as blue, hydrogen as yellow, natural gas as red, and liquid fuels as gray. Primary biomass is shown as green, captured $CO_2$ is shown as purple, fossil primary energy (oil and natural gas) is shown as black, and primary carbon-free energy (wind, solar, nuclear, and hydro) is shown as light blue. Electricity and natural gas inputs to the captured $CO_2$ box are energy inputs for direct air capture (DAC).

**Table 1 | Markets that are forced to clear (supply = demand) in the simple energy system model**

| Sector | Supply (Production Technologies) | Demand (Final or Intermediate Consumption) |
|---|---|---|
| Electricity | NG, NG w/CCS, Bio, BECCS, CFE, $H_2$ | Final demand, hydrogen, liquids, NG, DAC |
| Hydrogen | NG, NG w/CCS, Bio, BECCS, electrolysis | Final demand, electricity, liquids, NG |
| Liquids | Oil, Bio, BECCS, synthetic fuel | Final demand |
| Natural gas | Fossil NG, Bio, BECCS, synthetic fuel | Final demand, electricity, hydrogen, DAC |
| Biomass | Biomass primary energy | Final demand, electricity, hydrogen, liquids, NG |
| Captured $CO_2$ | Energy with $CO_2$ capture, DAC | Utilization (synthetic fuel), geologic sequestration |

Each row in the table is associated with a constraint in the least-cost optimization. CFE refers to Carbon-Free Electricity (wind, solar, hydro, and nuclear). NG refers to Natural Gas. BECCS refers to Bioenergy with Carbon Capture and Sequestration, although the disposition of $CO_2$ between sequestration and utilization is a model choice. DAC refers to Direct Air Capture.

performance assumptions described in Methods. The model solves by minimizing total cost in the single year represented.

Figure 2 shows the markets represented, and Table 1 shows the supply and demand possibilities within these markets. Within electricity, at least 9% of total generation is constrained to come from sources other than wind, solar, nuclear, and hydro, consistent with the median share of electricity from these sources in the 1.5 °C scenarios in the IPCC AR6 database[28]. A sensitivity case in which this constraint is removed is shown in Supplementary Fig. 4. Gross $CO_2$ emissions from remaining fossil fuels minus the amount of $CO_2$ stored geologically, is specified to be zero for a net-zero energy $CO_2$ system.

In a net-zero $CO_2$ system, land use change emissions related to bioenergy production are appropriate to include but are assumed to be zero in our core cases, with positive values considered in a sensitivity case (Supplementary Fig. 3; see further discussion in Methods). A central estimate near zero is appropriate when a large share of the biomass supply is assumed to come from waste streams and second-generation feedstocks[40–42], which is generally observed in strong mitigation scenarios[35,43]. Regarding non-$CO_2$ emissions, in the IPCC 1.5 °C scenarios, while energy-related $CO_2$ emissions are typically near zero in 2050 (Supplementary Table 1), non-$CO_2$ forcing is still positive when net-zero $CO_2$ emissions are attained, meaning that these scenarios do not attain net-zero GHG emissions until well after 2050[1]. For this reason, we do not include upstream emissions factors from fossil

fuels in the core cases but include them in a sensitivity case (Supplementary Fig. 3; see further discussion in Methods). The main findings of this study are robust to choices about these upstream emissions factors (compare Supplementary Figs. 1 and 3).

To examine the roles of biomass supply and $CO_2$ sequestration in driving key outcomes, we vary constraints on these resources independently, as well as jointly. These modeling constraints are meant to reflect differences in how society may choose to use or limit these resources and do not primarily reflect judgments about physical resource availability such as land available to grow biomass or underground pore space available for $CO_2$ storage. Specifically, four core cases are examined: (1) unconstrained biomass supply and $CO_2$ sequestration (UC); (2) biomass supply constrained (BC); (3) geologic $CO_2$ sequestration constrained (SC); and (4) $CO_2$ sequestration and biomass constrained (SBC). When biomass is constrained, total biomass primary energy is forced to be less than 64 EJ (in 2050), which is the 10th percentile from the distribution for the 1.5 °C (C1) scenarios in the IPCC AR6 database. This is considerably lower than estimates of global sustainable bioenergy potential reported by others[44,45].

Similarly, when $CO_2$ sequestration is constrained, total $CO_2$ sequestration is forced to be less than 3.8 Gt $CO_2$ per year (in 2050), which is the 10th percentile from the distribution for this variable. Although fossil energy can be retained to some extent in all scenarios that allow geologic $CO_2$ sequestration, the share of fossil energy

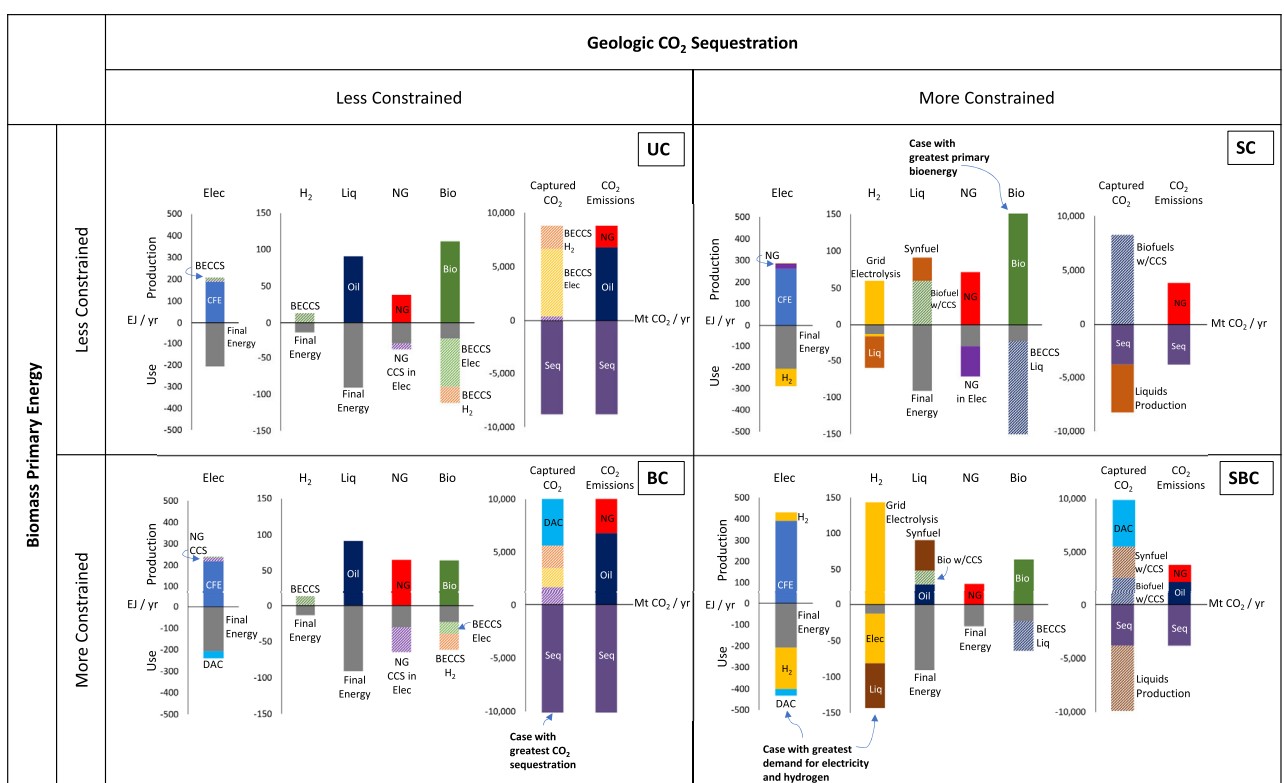

**Fig. 3 | Summary of the four illustrative net-zero energy systems.** In each panel, the first four bars show production (by technology) and use (by sector and technology) of electricity, hydrogen, liquids, and natural gas. The fifth and sixth bars show biomass primary energy and captured $CO_2$ (by source), respectively, along with their final dispositions. The final bar shows gross positive and negative $CO_2$ emissions. Final energy demands for each energy vector (electricity, hydrogen, liquids, natural gas, and biomass) do not vary across scenarios and are shown in gray (below the x-axis). Values below the x-axis in the first six bars represent the disposition of energy or captured $CO_2$. UC Unconstrained case, BC Biomass Constrained case, SC Sequestration Constrained case, SBC Sequestration and Biomass Constrained case. Supplementary Fig. 1 and Supplementary Tables 8–11 show the same results in alternative formats. CFE Carbon-Free Electricity. BECCS Bioenergy with Carbon Capture and Sequestration. DAC Direct Air Capture. Source data are provided as a Source Data File.

necessarily declines as the geologic sequestration constraint is tightened. In the limit of zero $CO_2$ sequestration, the resulting system would approach 100% carbon-free energy, or 100% renewable energy (if nuclear is also disallowed). Such systems have been reviewed elsewhere[46]. The lower amount used in this study (3.8 Gt per year) is lower than the amount deployed in the Low Investable Storage Potential 1.5 °C case considered by Grant et al.[30]. In all cases considered here, the sources of captured $CO_2$ (fossil, biomass, DAC) are determined endogenously by the optimization.

In scenarios in which these variables (biomass supply and geologic $CO_2$ sequestration) are unconstrained, their values are determined endogenously by the optimization. In all four cases, we assume the same final energy demand for electricity, hydrogen, liquid fuels, natural gas, and biomass (median values from IPCC 1.5 °C scenarios, shown in Supplementary Table 2), as well as the same constraint on the share of carbon-free (wind, solar, hydro, and nuclear) sources in electricity (<91%) and the same net energy-related $CO_2$ emissions (zero).

Although the simple model tends to produce rather stark differences between net-zero cases—an expected result from linear optimization in a single region and period—the results clarify the linkages between key constraints and outcomes. Figure 3 shows production and use of the four major energy carriers represented in the model—electricity, hydrogen, liquids, and natural gas—as well as production and final disposition of primary bioenergy and captured $CO_2$. In addition, gross positive emissions from fossil energy and gross negative emissions from sequestration are shown.

In the unconstrained (UC) and biomass constrained (BC) cases, remaining liquids (91 EJ) are conventional oil-derived fuels. Emissions are offset by removals (CDR) in the form of either BECCS in hydrogen and/or electricity (in both the UC and BC cases) or DACCS (in the BC case only). In these scenarios, CDR is selected because it is less costly than replacing conventional fuels with biofuels or other synthetic fuels. BECCS accounts for all CDR when biomass is unconstrained (UC), whereas DACCS accounts for a significant share of CDR when biomass is constrained (BC). BECCS deploys preferentially in the unconstrained case because it is more cost-effective than DACCS, given the capital and energy requirements associated with DACCS.

While BECCS could be deployed in any sector, BECCS deploys preferentially in hydrogen and electricity generation under our default assumptions when there are no $CO_2$ sequestration constraints. This preference arises, in part, because nearly all carbon in the biomass can be captured when the final product (electricity or hydrogen) does not contain carbon, thereby increasing the effective (negative) carbon intensity of these production technologies. Since BECCS electricity and BECCS hydrogen provide similar functions in a net-zero energy system (CDR), their relative deployment is determined primarily by cost differences. BECCS hydrogen is slightly more cost-effective under our default assumptions, leading it to deploy up until the demand for hydrogen is satisfied, followed by deployment of BECCS electricity. Other assumptions may result in a different cost ordering.

When only $CO_2$ sequestration is constrained (SC), the amount of allowable $CO_2$ storage is not sufficient to offset all emissions from remaining liquid fuels and natural gas if they are produced using conventional fossil sources, so biofuels displace a share of conventional liquids[47]. Applying CCS to biofuels production (i.e., deploying BECCS in liquids rather than biofuels without CCS) is preferable because this option provides CDR simultaneously, offsetting remaining emissions from fossil natural gas. Once the imposed $CO_2$

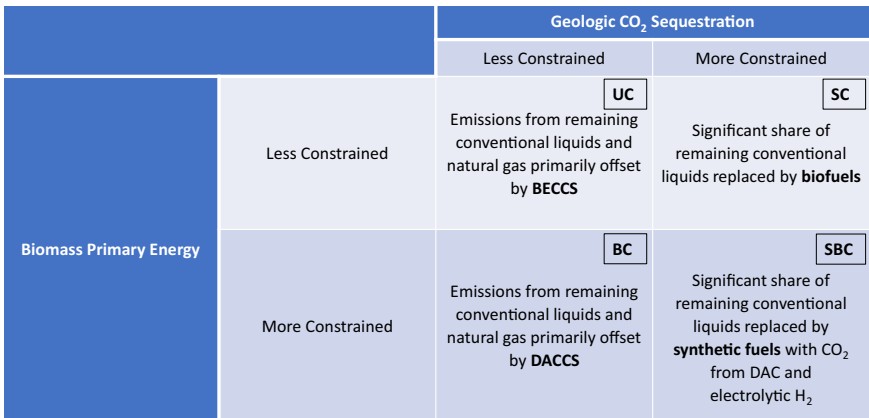

**Fig. 4 | Different routes to mitigate emissions from remaining liquids and natural gas in net-zero scenarios.** These routes are abstracted from the results of the illustrative core cases discussed above to highlight the dominant feature of each quadrant. Modeled scenarios—whether from the simple model or more complex models—will typically combine elements from different quadrants, even if one option is dominant. *UC* Unconstrained case, *BC* Biomass Constrained case, *SC* Sequestration Constrained case, *SBC* Sequestration and Biomass Constrained case. BECCS refers to Bioenergy with Carbon Capture and Sequestration, DAC refers to Direct Air Capture, and DACCS refers to DAC with Carbon Sequestration.

sequestration limit is reached, the model can either deploy biofuels without CCS or synthesize fuels from hydrogen and captured $CO_2$ to satisfy remaining liquids demand. The latter option is more cost-effective if the additional $CO_2$ is derived from biofuels with CCS (rather than from DAC) because the incremental cost of $CO_2$ capture on biofuels production is relatively small. Conceptually, this route is similar to recycling more $CO_2$ and adding supplemental hydrogen within the biofuels production process[48,49]. In addition, there is a greater preference for synthetic fuels when more emissions are assumed to be associated with bioenergy production (compare Supplementary Figs. 1 and 3). When both $CO_2$ sequestration and biomass are constrained (SBC), the share of liquids from the synthetic route increases, since biomass supply is constrained, with a greater share of the captured $CO_2$ coming from DAC.

A few general insights emerge from these illustrative scenarios. First, although biomass is not forced to utilize CCS, biomass is typically coupled with CCS in mitigation scenarios because of the significant economic value associated with $CO_2$ removal[50]. However, the sectoral allocation of biomass varies between cases. Biomass preferentially deploys as CDR when $CO_2$ sequestration is unconstrained (UC or BC) but is used primarily to make biofuels as a substitute for conventional liquids when $CO_2$ sequestration is constrained (SC or SBC) because not all emissions from remaining liquids fuels can be offset. While BECCS deploys in electricity or hydrogen when used for CDR in our core cases, it deploys in liquids production when the costs of BECCS electricity and BECCS $H_2$ are assumed to be higher, even when $CO_2$ sequestration is unconstrained (Supplementary Fig. 6). In this case, however, most of the remaining fuels-related mitigation still comes from CDR (BECCS) to offset fossil emissions rather than from biofuels as a substitute for conventional fuels (see below). Second, DAC and synthesized fuels can deploy together (SBC), but neither is necessary for the other. Specifically, DACCS can provide CDR when biomass is limited (BC), and synthetic fuels can utilize $CO_2$ from biomass (biofuels with CCS) when $CO_2$ sequestration is limited (SC).

Two other insights pertain to the production of molecular fuels. First, the scale of hydrogen production varies more than other energy carriers (by more than an order of magnitude) across the illustrative cases. Greater hydrogen production occurs when other sources (biomass, natural gas) are more constrained for use in electricity or when hydrogen is needed to synthesize fuels. Second, fossil natural gas is largely retained in our core cases, whereas conventional liquid fuels are highly dependent on the biomass and $CO_2$ sequestration assumptions. When there is limited ability to offset emissions, it is less costly to

displace fossil liquid fuels than fossil natural gas, due in part to the relative differences in carbon intensities and base costs of these fuels, with oil assumed to be considerably more expensive than fossil natural gas under our default assumptions. In other words, because the price of fossil natural gas and its carbon intensity is lower than oil, the abatement cost for an alternative natural gas production technology is higher than it is for an alternative liquids production technology, all else equal. However, when the price spread between oil and fossil natural gas is assumed to be narrower, most fossil natural gas is displaced in the SC and SBC cases (Supplementary Fig. 5). It is also worth noting that the assumed exogenous final demand for natural gas is considerably lower than the final demand for liquid fuels (30 EJ for natural gas versus 91 EJ for liquid fuels), although there is significant variability across IPCC 1.5 °C scenarios in 2050 (see Supplementary Table 2).

## Drivers of emissions mitigation from remaining fuels

Figure 4 qualitatively summarizes how assumptions about biomass supply and $CO_2$ sequestration drive differences in mitigation outcomes related to remaining liquid fuels and natural gas. When $CO_2$ sequestration is less constrained, emissions from remaining conventional fuels tend to be offset by CDR, which preferentially takes the form of BECCS when biomass supply is less constrained and DACCS when biomass supply is more constrained. When $CO_2$ sequestration is more constrained, more alternative fuels are deployed, primarily taking the form of biofuels when biomass is less constrained and synthetic fuels when biomass is more constrained. These four routes are not mutually exclusive, with the amount of mixing determined by the stringency of the constraints. As the biomass constraint is varied (from more restrictive to less restrictive), we find that the share of biomass-based solutions (BECCS for CDR and/or biofuels) increases monotonically. Similarly, as the $CO_2$ sequestration constraint is varied (from more restrictive to less restrictive), the share of CDR in emissions mitigation from remaining fuels increases monotonically.

To evaluate the robustness of this outcome, the decomposition from Fig. 1 can be applied to the four illustrative cases and the sets of sensitivity cases defined in Supplementary Table 13. In Fig. 5, the core cases are shown along with sensitivities examining lower demand, higher assumed GHG intensities for fossil fuels and biomass, no restriction on the amount of carbon-free electricity (CFE), alternative fuel prices, and higher BECCS costs. The unconstrained (UC) and biomass constrained (BC) cases rely heavily on CDR, so the CDR share of remaining fuels-related mitigation is high in all cases and often accounts for all fuels-related mitigation, with the primary difference

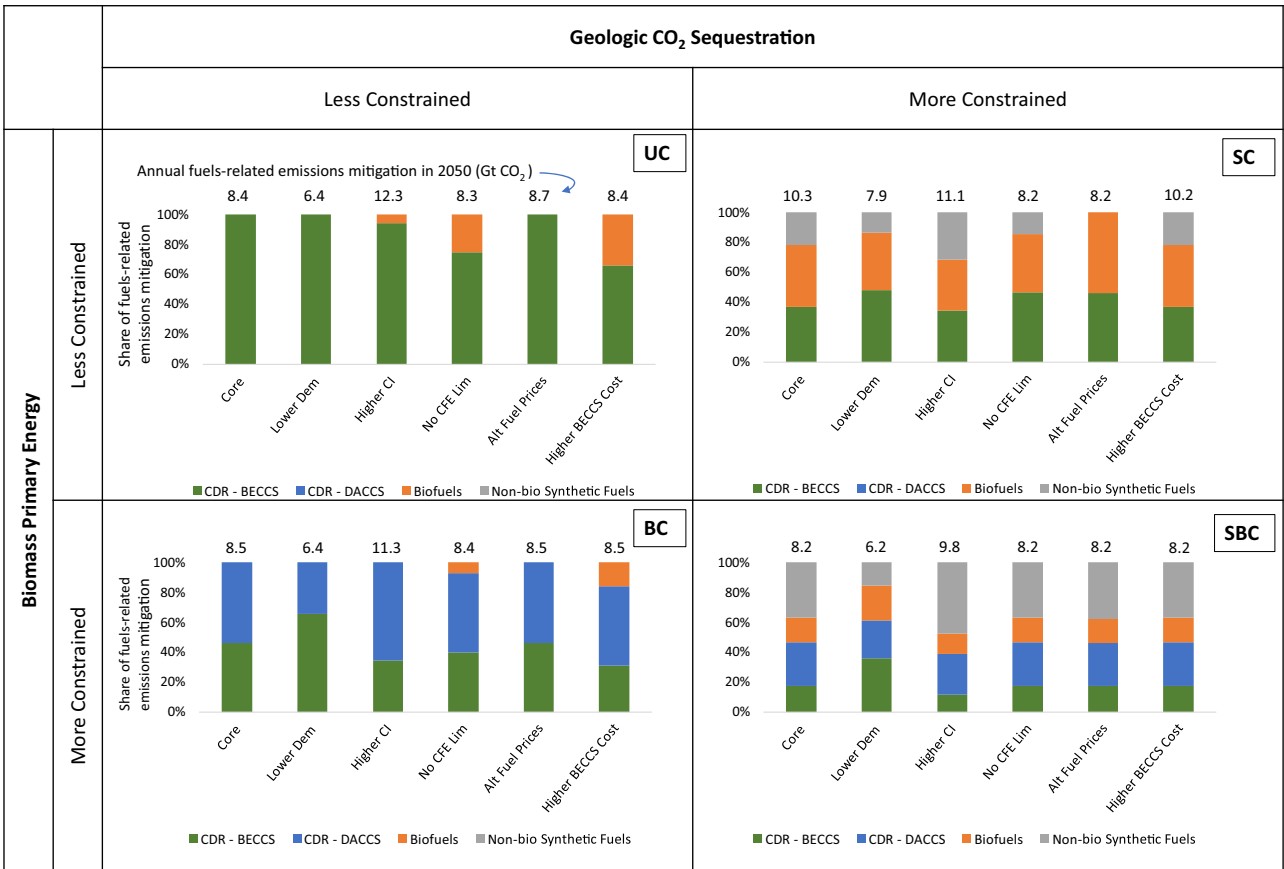

**Fig. 5 | Mitigation decomposition for remaining liquid fuels and natural gas in 2050 from the core and sensitivity scenarios.** Sensitivity scenarios are defined in Supplementary Table 13, and additional information related to this chart is provided in Supplementary Table 15. *UC* Unconstrained case, *BC* Biomass Constrained case, *SC* Sequestration Constrained case, *SBC* Sequestration and Biomass Constrained case. BECCS refers to Bioenergy with Carbon Capture and Sequestration, and DACCS refers to Direct Air Carbon Capture and Sequestration. Source data are provided as a Source Data file.

between UC and BC being the share of BECCS versus DACCS. In the CO$_2$ sequestration constrained (SC) cases, CDR, biofuels, and synthetic fuels all contribute to emissions mitigation from remaining fuels, with biofuels accounting for more than a third of total mitigation in all cases. In the SBC case, in which CO$_2$ sequestration and biomass are both limited, synthetic fuels provide more mitigation than biofuels in five of the six cases shown.

When biomass supply and CO$_2$ sequestration are both forced to be limited (as in the SBC case), the production of synthetic fuels requires a significant increase in hydrogen production. Increased hydrogen production, in turn, leads to an increase in electricity generation, effectively defining a reciprocal relationship between the use of these resources and the size of the electricity system. When carbon-free electricity is unconstrained, the growth in electricity and hydrogen use is less extreme as there is less direct coupling of these sectors, but the general result holds (compare Supplementary Fig. 1 and 4).

Moreover, biomass supply and geologic CO$_2$ sequestration are themselves inversely related. When CO$_2$ sequestration (and therefore CDR) is limited (SC), biofuel deployment increases as more conventional fossil liquids must be replaced with alternative fuels. Although less CDR might reduce bioenergy needed for BECCS, more primary bioenergy is used to produce biofuels than would be used by BECCS to offset the associated emissions from an equivalent amount of conventional fuels. For example, if the carbon intensities (per unit energy) of oil and biomass are comparable, then 2 EJ of primary bioenergy are needed to replace 1 EJ of oil-based liquid fuel (assuming a conversion efficiency of ~50% to produce fuel from biomass), whereas only 1 EJ of primary bioenergy would be needed in the form of BECCS to offset the

emissions from 1 EJ of oil-based liquid fuel, assuming all of the CO$_2$ from BECCS could be captured and sequestered, which is approximately true for BECCS in power or hydrogen production.

Conversely, when biomass supply is limited, use of DACCS expands. Since DACCS consumes rather than produces energy, it does not displace other fossil energy technologies as BECCS would, thereby increasing the need for CCS directly (on the fossil technologies) or indirectly (by requiring more DACCS to offset unabated fossil emissions). In either case, we find that CO$_2$ sequestration increases as biomass is reduced.

An important implication of these findings is that limiting one resource (for example, biomass) may inadvertently put pressure on another (for example, CO$_2$ sequestration). While both biomass and CO$_2$ sequestration could be jointly limited, this would put additional pressure on the electricity system. The consequences of these choices are not small. For example, the SBC case requires a ~5x increase in electricity production by 2050 relative to current production, essentially doubling what is needed to realize the assumed ~2.5 x increase in electricity final demand due to end use electrification.

These outcomes suggest important differences in the overall scale of resources required across the cases, which are also reflected to some extent in differences in the marginal abatement costs. For example, the marginal abatement cost of SBC is over four times larger than the cost of UC (Supplementary Table 12 and Supplementary Tables 16–20). Even when overall final demand is considerably lower because of demand-side mitigation, electricity production in the SBC case would still expand by ~3.5 x relative to today (see Supplementary Fig. 2). These reciprocal relationships—between CO$_2$ sequestration and

biomass supply, as well as between these resources and electricity generation—are robust to several key assumptions (see Supplementary Table 14 and Supplementary Fig. 2-6), including assumptions about final demand. In the most general sense, these relationships hold whenever the constraints on biomass and $CO_2$ sequestration are binding.

## Societal tradeoffs

The relationships between different resources and technologies explored in the illustrative systems highlight alternative transition pathways and end states, with different, albeit uncertain implications for a wide range of societal outcomes. Early consideration of these potential outcomes may affect preferences and ultimately choices about alternative pathways. For example, regional air quality, land and water use, biodiversity, food prices, manufacturing supply chains, critical materials requirements, and employment composition, to name a few, could be affected by both the type and location of energy system expansion. How these outcomes are evaluated may vary considerably by region. Societal preferences may also depend, in part, on how energy security and system resilience are perceived to be affected, and differences in overall cost could affect decisions to the extent that costs are borne by consumers or limit the resources for other societal priorities. Among the cases considered here, the case with unconstrained bioenergy and $CO_2$ sequestration (UC) is the least costly since the combination of these resources enable lower-cost CDR in the form of BECCS.

Among the three constrained cases (BC, SC, SBC), BC uses the least amount of biomass and electricity, but it relies heavily on geologic $CO_2$ sequestration. Given the biomass and electricity outcomes, the land use and water use requirements in this case are potentially smaller than in the other constrained scenarios. However, given that land and water implications will depend on more granular technology decisions than what we are able to represent, a quantitative assessment is outside the scope of what can be reasonably accomplished here. On the other hand, this case would require broad acceptance of CDR[51–54]. It further presumes a stable source of revenue for negative emission technologies, which typically do not derive a significant share of revenue from traditional energy markets and are therefore likely to be reliant on policy support. The role of nature-based CDR, which would affect the call on geologic sequestration, has not been considered here, but the general point about CDR acceptance and policy support remains.

SC relies less on $CO_2$ sequestration than BC, but it deploys more total electricity and relies most heavily on biomass. In addition to the potential water, land use, and agricultural commodity price implications of bioenergy[55–60], similar questions may arise with respect to electricity infrastructure expansion[61–63]. Therefore, this pathway most directly forces questions about the role of land in climate change mitigation. Beyond the demand for land related to bioenergy and electricity expansion, land may be deployed for natural climate solutions[64], other ecosystem services[65], and growing food and timber needs. How these competing claims on land are perceived and adjudicated could have profound implications for the future role of bioenergy.

SBC is comparable to SC in terms of $CO_2$ sequestration and comparable to BC in terms of biomass, but as a result, its electricity production is largest. In this case, electricity not only serves growing final energy demand from end uses as it does in the other cases but is also used to produce more than an order of magnitude more hydrogen than what is produced in the BC case. This additional hydrogen, in turn, is combined with captured $CO_2$ to produce a significant share of total liquids and is used in the power system when use of natural gas would be more challenging due to the sequestration limit. The expansion of electricity and hydrogen infrastructure in this case potentially shifts the source of land pressure to these sectors[62,66]. In addition, except for remaining natural gas, all energy carriers in this case are derived directly or indirectly from electricity. While lowering final demand through energy efficiency, mode switching and/or conservation could alleviate some of the pressure on the electricity system, it is unlikely to fully eliminate it. SBC is the costliest of the four cases examined, because non-biomass synthetic fuels are deployed as the marginal abatement technology.

Many net-zero scenarios assessed by the IPCC and produced by energy system models have relied on CDR to mitigate emissions from remaining liquid fuels and natural gas. Storylines characterized by supply-side resource limitations (or preferences for less use of such resources) have been explored through low-demand scenarios[67,68]. However, the sensitivity cases discussed above and results from other studies[69] suggest that the tradeoffs discussed here still exist under scenarios with lower final demand.

## Discussion

In this paper, we have used a simple structural model of the global energy system to develop several illustrative net-zero energy systems. We find that for a given set of final energy demands, assumptions about the amount of available biomass supply and $CO_2$ sequestration, reflecting regional preferences about the use of these resources, drive key differences in how emissions from remaining fuels are mitigated. Net-zero systems with less constrained carbon dioxide ($CO_2$) sequestration use more CDR, whereas systems with more constrained $CO_2$ sequestration deploy more low-carbon fuels. Systems with more biomass rely on bioenergy with carbon capture and storage (for CDR) or biofuels, whereas systems with more limited biomass supply rely more heavily on direct air capture for CDR or synthetic fuels. We find that limiting one resource (e.g., biomass) often increases reliance on another (e.g., $CO_2$ sequestration). These interactions are important because constraints on these resources in scenarios likely reflect a range of factors, including views about their ancillary or indirect impacts. The key insights from these cases are robust to assumptions about final energy demands, emissions factors, fuel prices, and technology costs, as shown in Fig. 5 and Supplementary Figs. 1–6.

The integrated assessment and energy system modeling communities have a demonstrated record in supporting climate and energy-related decision-making in multiple areas[12,70]. Going forward, a shift in focus toward fuels decarbonization may further enable this community to inform emerging choices related to the energy transition. This shift in emphasis is a natural progression, as MES model development has been motivated, in part, by the opportunity to extend power sector capacity expansion approaches to other sectors[6,18]. In addition, many of the early lessons from power sector modeling have already been incorporated into integrated assessment[71], which can be seen to some extent in the broad agreement in power sector outcomes (Supplementary Table 7). These lessons are also partially reflected in the domain of policymaking, where recent U.S. policies, such as the Inflation Reduction Act are projected to deliver significant power sector mitigation[72], commonly found to be the lowest-cost source of $CO_2$ emissions abatement in models[73].

Despite progress in other areas, the transition path for remaining liquid fuels and natural gas is not currently characterized by an easily identifiable consensus. Although there is apparent agreement about the importance of CDR in scenarios assessed by the IPCC[1], any apparent model agreement related to emissions mitigation from remaining fuels could be spurious, given the relative lack of alternative technologies included in many models. Even when such technologies are represented, differences may emerge in the parameterization of complex liquids production processes, as there is relatively sparse literature connecting detailed process modeling to the types of inputs needed for energy system modeling. These gaps, if they persist, may hinder applicability to policy, planning, and investment decision-making, particularly if policy interest tilts toward fuels decarbonization in pursuit of deep decarbonization.

With multiple modeling approaches, established institutions for assessing and comparing models, and a growing research focus on net-zero systems[74] and scenarios[26], the modeling community is well positioned to continue providing actionable information, particularly if key steps are taken to anticipate future needs. First, existing models should be refined to represent all foreseeably relevant technologies[75–77]. To avoid apples-to-oranges comparisons, models that do not include most major categories of CDR and alternative fuel production technologies should be modified to include them. This development could be facilitated by expansion of process-level modeling of key emerging technologies with this application in mind. Inclusion of all major technologies using informed parameterizations enables consideration of the widest possible solution space and facilitates more useful comparison between outcomes. Even when not all technologies deploy, their inclusion avoids speculation about whether differences in outputs are primarily due to differences in technology availability or parameterization.

Second, the modeling capability set should be expanded. The availability of a suite of established IAMs has facilitated model comparison, assessment of the variability in model outcomes, and enabled consensus conclusions to be developed in the context of the IPCC[1]. These lessons suggest that development of a larger suite of MES models could be useful for strengthening the impact of this community[14]. Such development could provide a wider set of complementary and timely insights given the ability of MES models to represent higher resolution aspects of the energy system, including the coupling between the electricity sector and other energy sectors, as well as transmission and energy storage expansion, all of which could provide more granular information for decision-makers while simultaneously informing the parameterization of coarser-resolution models.

Third, the scenario space considered in net-zero modeling should be enlarged. Given the importance of key constraints such as those related to biomass supply and $CO_2$ sequestration, which may reflect societal preferences more than physical resource availability, these constraints—and other factors likely to explain differences—could be explored more systematically across different models and frameworks. More generally, enlarging the scenario space considered by both IAM and MES studies would be useful to pinpoint and highlight major differences between alternative transition pathways. These differences are particularly important to examine when they involve highly uncertain aspects of the energy transition or aspects that are most consequential for other societal priorities, as these choices will presumably require the most attention by decisionmakers. This discussion is arguably a prerequisite for effective policymaking, given the diverse array of stakeholder and national interests at play, and would ultimately help to shape investment priorities, including R&D priorities.

Finally, opportunities should be sought to combine the complementary strengths of different types of models and frameworks[78], recognizing the different underlying disciplines represented in MES and integrated assessment. For example, IAMs could provide information about land-related $CO_2$ and non-$CO_2$ forcing responses, as well as about responses in energy end uses that could be useful inputs to MES models. MES models, in turn, could provide more fidelity in supply and conversion sectors, with spatial and temporal resolution (including transmission and storage detail) necessary to credibly project outcomes at a sub-national level. MES models may therefore clarify when higher resolution is (or is not) needed to project aggregate outcomes, which in turn, could help prioritize aspects of IAM development. Both types of models can inform priorities for engineering process models by specifying the use cases for certain processes, which can then be used to improve the specification of such technologies. Ultimately, a combination of approaches could be used to evaluate broader consequences and tradeoffs across alternative pathways to net-zero, ideally informing, in a more comprehensive way, the various choices that will collectively define the energy transition.

## Methods

### Energy supply and demand

For each final energy carrier (electricity, hydrogen, liquid fuels, natural gas, and biomass), the exogenous final energy demand is set equal to the 2050 median (P50) value from the 97 C1 scenarios in the IPCC AR6 database[1,79]. Supplementary Table 2 shows the median values, along with other percentiles from the distribution for each variable. A sensitivity case with lower final demand assumptions is considered in Supplementary Fig. 2. Coal is not included as a final energy demand in the simple model because its consumption is small relative to the other final energy demands in most scenarios at the time of net-zero emissions.

For each energy carrier represented in the model, the technology options shown in Table 1 in the main text compete based on cost, subject to the other constraints in the model. In general, we have sought to characterize key technology classes that would be most relevant for illustrating the type of energy transition choices examined in this paper, but in the spirit of developing a transparent structural model, we have not attempted to comprehensively represent all potential process configurations for a given technology class. Furthermore, given regional, temporal and process heterogeneity that cannot be represented in this simple framework, all assumptions should be considered illustrative.

The cost of each technology is assumed to consist of a non-energy component (capital and O&M) and an energy (fuel) component. The non-energy components are specified exogenously for each technology. The energy component may consist of the costs of energy carriers whose prices are endogenous. To simplify the modeling, the objective function is the sum of non-energy costs for all technologies and the costs of primary energy (oil, fossil natural gas, and biomass) for technologies that consume them. For a given technology, the energy inputs —and therefore energy costs—will depend on the conversion efficiency, which is also specified exogenously. Unless otherwise stated, each technology is assumed to have only one energy input. Non-energy costs and conversion efficiencies for the technologies represented are shown in Supplementary Tables 3–6. Levelized costs are estimated using an 85% capacity factor and a 13% capital recovery factor unless otherwise stated.

The levelized non-energy cost for direct air capture (DAC) is assumed to be $200/ton following central estimates from several studies[80–82]. These assumptions are broadly consistent with non-energy cost estimates for mature sorbent and solvent systems from NETL[83,84]. Two processes are represented in our framework, one that takes electricity as an input and one that takes natural gas as an input. In both cases, 0.147 tons $CO_2$ are assumed to be captured per GJ of energy input.

In general, consistency regarding technological evolution is desirable. When possible, we have sought to use cost and performance information for the year 2050, which would consistently consider evolution in technology, to the extent that the underlying sources are consistent in this regard. In some cases, costs and conversion efficiencies may be estimated to be quite different than today's values, while in other cases the technology evolution may be more gradual, depending in part on underlying characteristics of the technology. Where values for 2050 are not available, we have typically used long-term or Nth-of-a-kind estimates.

### Fuel prices

Fuel prices are exogenous and therefore unresponsive to changes in demand. The default price of crude oil is assumed to be $20 per GJ, the price of conventional (fossil) natural gas is assumed to be $6 per GJ, and the price of biomass is assumed to be $10 per GJ. Because fuel prices will vary significantly by region and over time, we have included a sensitivity with higher prices for natural gas and lower prices for bioenergy (Supplementary Fig. 5). In selecting assumptions for prices, we consider that, in net-zero scenarios, natural gas demand and

therefore prices are likely to be lower, and conversely that biomass demand and prices are likely to be higher than they would be in less stringent scenarios.

As an example, the IEA[31] states that in its net-zero scenario "prices are increasingly set by the operating costs of the marginal project required to meet demand, and this results in significantly lower fossil fuel prices than in recent years." Table 2.1 in the IEA Net-Zero by 2050 study[31] shows natural gas prices in 2050 converging across regions at values below $6 per MMBtu, which is approximately our default assumption. That said, as a general matter, it is interesting to consider a case in which the spread between gas and oil prices is smaller, because, among other things, this could change whether liquids or natural gas emissions are mitigated first. This can be accomplished by raising the NG price and we have used $15 per GJ for this sensitivity[85].

Regarding biomass prices, the EMF-33 study is one of the more recent multi-model efforts that has examined biomass supply. Rose et al.[43] show supply curves for bioenergy up to several hundred EJ in 2050. That study evaluated global biomass supplies at prices ranging from $3 to $15 per GJ in 2005 dollars (~$5 to $23 per GJ in current dollars) and found that 150-200 EJ/year is available at prices ranging from $3 to $8 per GJ in 2005 dollars (~$5 to $12 per GJ in current dollars). As another point of comparison, the EPRI results in Fig. 1[35] estimate biomass market prices from ~$9 to $29/GJ for 2050 across its three U.S. net-zero by 2050 scenarios. These sources support a default biomass price assumption around $10 per GJ. That said, as a sensitivity, we have used $5 per GJ, consistent with the lower end from EMF-33.

## CCS capture rates

For CCS technologies, the capture fraction is assumed to be 95%. While 90% has been a common assumption in the literature, that is not a technical limit. Given strong incentives to reduce remaining emissions in net-zero scenarios such as those considered in this study, higher rates could be cost-effective. While it is not possible in our study to optimize for the capture rate, several studies have assumed or assessed the possibility of higher capture rates, in some cases approaching 100% capture[86–88].

Brandl et al.[87] specifically states "in no case, was a capture rate of 90% found to be optimal, with capture rates of up to 98% possible at a relatively low marginal cost," and Du et al.[88] states that "power plants can achieve zero-emissions with CCS at competitive costs". In our study, we have chosen 95% to be in between the more historically common but conservative 90% assumption and the more scenario-consistent but ambitious near-100% assumption. It is worth noting that NETL assumes 95% capture in its recent baseline study on CCS[89].

For all technologies, given the same capture rate assumption, the resulting emissions capture coefficient (tons $CO_2$ captured per GJ out) varies by technology given differences in the carbon content of fuels as well as differences in conversion efficiencies. For BECCS, there are large differences between technologies. We estimate the emissions capture coefficient as the capture fraction multiplied by ($CI_{bio}/eff - CI_{carrier}$). $CI_{bio}$ is the carbon intensity of primary bioenergy (assumed be 0.1 tons per GJ). $eff$ is the conversion efficiency of the process, which is lowest for BECCS in electricity (22%) and highest for biogas production with CCS (60%). $CI_{carrier}$ is the carbon intensity of the energy carrier produced by the process (0 for electricity and hydrogen; equal to the carbon intensity of conventional liquids and natural gas in the case of BECCS coupled with liquids production or BECCS coupled with gas production, respectively). The resulting carbon capture coefficients (which can be interpreted as negative emissions coefficients if the $CO_2$ is sent to storage) are 0.48, 0.17, 0.15, and 0.11 tons per GJ for BECCS in electricity, hydrogen, liquid fuel production, and gas production, respectively. Each of these is reported on an output basis (per GJ of energy produced). The cost of geologic sequestration is assumed to be $10 per ton $CO_2$, not including the cost of capture, which is included in the relevant non-energy costs shown in Supplementary Tables 3–6.

## Energy feedstock emissions

Energy feedstock emissions are handled differently depending on where they are assumed to occur. Combustion-related feedstock $CO_2$ emissions related to other aspects of the life cycle inside the energy system boundary—for example, within transportation or industry—are implicitly included in the analysis to the extent that the corresponding energy demands are included in the exogenous final energy demands, which span transportation, industry, and buildings. Regarding bioenergy, land use change emissions related to bioenergy production are appropriate to include but are assumed to be zero in our core cases, with positive values considered in a sensitivity case. A central estimate near zero is appropriate when a large share of the biomass supply is assumed to come from waste streams and second-generation feedstocks[40–42], which is generally observed in strong mitigation scenarios[35,43]. Negative land use change emissions are also possible and imply net carbon sequestration on land without the use of CCS (BECCS). Negative land use change emissions (net carbon sequestration) can occur when the land use and management changes associated with increasing biomass production result in increases in below and above ground carbon stocks that more than offset the decreases in carbon stocks from any land displaced directly or indirectly by such production. Given variability in these estimates around zero for second-generation biomass crops[40–42,90], zero is used as a central assumption in our core cases, but we examine a sensitivity case with a higher assumed emission factor for bioenergy production (discussed below).

While $CO_2$ emissions across the life cycle are relevant to specifying emissions in a net-zero $CO_2$ system, the inclusion of non-$CO_2$ emissions is effectively a question about scenario design. In the IPCC 1.5 °C scenarios, while energy-related $CO_2$ emissions are typically near zero in 2050 (Supplementary Table 1), non-$CO_2$ forcing is still positive when net-$CO_2$ emissions are attained, meaning that these scenarios do not attain net-zero GHG emissions until well after 2050[1]. Including all non-$CO_2$ emissions related to the energy system would shift the focus from a net-zero $CO_2$ system toward a net-zero GHG system. For these reasons, we do not include upstream emissions factors from fossil fuels in the core cases but include them in a sensitivity case (discussed below). It should be noted that energy-related non-$CO_2$ emissions factors could be considerably lower than today due to changes in production driven by mitigation incentives in stringent cases, such as the ones we consider. Thus, taken together, the two sets of assumptions about upstream emissions factors examine reasonable variation in the definition of a net-zero system, spanning approximately net-zero $CO_2$ and net-zero GHG energy systems.

Based on the considerations above, we examined a sensitivity case using emissions coefficients reflecting higher land use change $CO_2$ emissions for biomass primary energy (0.02 ton $CO_2$ per GJ) and upstream non-$CO_2$ emissions for oil and natural gas (0.006 ton $CO_2$e per GJ for oil and 0.014 ton $CO_2$ per GJ for fossil natural gas) as shown in Supplementary Fig. 3. These assumptions are based on assumptions used in the Open Energy Outlook[33]. These emissions coefficients arguably underestimate the mitigation of upstream emissions that would occur in a strong mitigation case and could therefore be viewed as a reasonable high-end case. Generally, we find that the case with higher emissions factors is more stringent, as it effectively requires net-zero GHG emissions, and therefore net-negative $CO_2$ emissions, rather than net-zero $CO_2$ emissions. However, the key findings discussed in this paper are robust to the choice of emissions factors (compare Supplementary Figs. 1 and 3).

## Carbon-free electricity share

Besides constraints associated with the key supply and demand balances shown in Table 1, three other constraints, along with the net emissions constraint, affect the solution. As discussed earlier, the constraints on bioenergy and $CO_2$ sequestration are based on the P10

values in the IPCC C1 scenarios in 2050. Another constraint restricts the CFE bundled technology (wind, solar, nuclear and hydro) from taking 100% market share in electricity production. The maximum market share is set equal to the median value of this share in the IPCC C1 scenarios. Supplementary Table 7 shows the distribution of the CFE share in IPCC C1 scenarios, as well as an alternative low-emissions share (LE share), which includes CCS. Wind, solar, nuclear, and hydro are bundled in the simple model because these four technologies are not represented explicitly in other sectors, so the share among them does not affect other outcomes. On the other hand, deployment of natural gas, hydrogen, and bioenergy deploy in multiple sectors. The impact of removing the constraint on CFE is shown in Supplementary Fig. 4.

### Robustness of model results to technology assumptions

While there are many different technology cost assumptions and it is not possible in this study to consider the sensitivity to each, we can consider which conclusions might be sensitive to specific assumptions. The UC and SC cases, which rely heavily on bioenergy, are likely to be sensitive to assumptions about bioenergy. The fuel price sensitivity discussed above considers lower cost biomass supply. The biomass constrained versions of these cases (BC and SBC) effectively consider higher biomass prices (via the shadow price on the biomass constraint).

Changes in the costs of specific bioenergy technologies could affect results in a more limited way. For example, changing the cost of BECCS $H_2$ versus BECCS electricity could change the cost and resulting deployment ordering of these technologies, although this would not affect the primary findings of this study. Raising the cost of BECCS in both power and $H_2$ production could move deployment of BECCS from these sectors to liquids or gas production. Because this could affect our conclusion about the specific roles of bioenergy across cases, we have considered a sensitivity case with BECCS electricity and BECCS $H_2$ non-energy costs increased by 50% relative to default values. Despite a shift in BECCS deployment from power and hydrogen to liquid fuels in the UC case under these assumptions, we find remaining fuels emissions are still primarily mitigated using CDR (BECCS) in this case (Fig. 5).

In the BC case, DACCS mitigates a significant share of remaining fuels emissions. The future cost of DACCS is quite uncertain. Since this case is biomass constrained, significantly increasing the non-energy cost of DACCS would not fundamentally change the solution, as there are no less expensive alternatives even with such higher DACCS costs. Similarly, significantly lowering the non-energy cost of DACCS would not change the solution in the UC case, unless DACCS were to become more cost-effective than BECCS.

In the SBC case, non-bio synthetic fuels play a more significant role, and the cost of producing such fuels is also quite uncertain. However, in this case, synthetic fuels deploy because other options (BECCS, DACCS, biofuels) are assumed not to be available. Therefore, raising the cost of producing synthetic fuels would increase the system cost at net-zero, but there is no lower-cost alternative that can be selected in this framework.

## Data availability

Source data are provided with this paper as a Source Data file.

## Code availability

The source code is available for this paper (https://doi.org/10.5281/zenodo.10709997).

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

## Acknowledgements

This work was authored in part by the National Renewable Energy Laboratory, operated by Alliance for Sustainable Energy, LLC, for the U.S. Department of Energy (DOE) under Contract No. DE-AC36-08GO28308. The NREL, PNNL and MIT authors acknowledge funding support from the ExxonMobil Technology and Engineering Company. The views presented in this paper are those of the authors alone and do not necessarily represent those of their institutions or funding entities. The U.S. Government retains and the publisher, by accepting the article for publication, acknowledges that the U.S. Government retains a non-exclusive, paid-up, irrevocable, worldwide license to publish or reproduce the published form of this work, or allow others to do so, for U.S. Government purposes.

## Author contributions

B.K.M. conceived the original idea, carried out the simulations, and took the lead in drafting the manuscript. H.J.H. and D.S.M. provided critical input and feedback on technology assumptions and contributed to the interpretation and visualization of model results. J.X.J., P.R.O., and A.V. helped to summarize and interpret data from other studies, provided critical feedback, and contributed to the interpretation and visualization of model results. S.K.R. provided critical input and feedback on resource assumptions, helped to summarize and interpret data from other studies, and contributed to the interpretation and visualization of model results. L.C., J.E., A.G., H.M., J.M., S.P., and D.C.S. contributed to the interpretation and visualization of model results, provided critical feedback, and helped to shape the final manuscript.

## Competing interests

The authors declare no competing interests.
