## [Peer Review File · Nature Communications]

REVIEWER COMMENTS

Reviewer #1 (Remarks to the Author):

Dear Editor, I have completed my review of the assigned manuscript. Based on the level of availability of CO₂ sequestration and biomass resources, the authors sought to investigate the drivers and implications of alternative routes to emission reduction from significant volumes of hydrocarbon fuels that remain in the energy system during net zero. The authors used a simple structural model of the global energy system based on exogenously specified final energy demands, biomass supply, and CO₂ sequestration. The work is interesting and relevant. However, some reported results would need further discussion and clarification for the benefit of prospective readers. Some minor corrections regarding typos and grammar are required in a few places as well. Please find my comments below, and I look forward to the revised version.

(1) The context in which 'fuel' in sentence 3 of the abstract is used is ambiguous.

Which types of fuels remain? For example, by the time net zero is reached, coal (which is a 'fuel') would have declined significantly. So, in such scenarios, 'significant coal volumes DO NOT remain even when net zero emissions are reached' compared to the levels today.

(2) CO₂ should be defined at first use.

(3) Introduction: paragraph 1.

“At the same time, significant amounts of liquid fuels and natural gas remain in most scenarios at the time of net-zero emissions”

It would help readers if some rationale or reason for this observation in such scenarios at the time of net zero emissions is given after this sentence. One may wonder why significant volumes of hydrocarbon liquid fuels may remain even after net zero. Authors could include reasons such as the energy requirements of hard-to-abate sectors and also high carbon lock-in in some countries/regions/sectors.

(4) “Another approach is to combine carbon that is removed from the atmosphere with hydrogen to make a synthetic (non-bio) hydrocarbon fuel (synthetic fuels)”

For clarity, it should be CO₂ with hydrogen instead of carbon with hydrogen. In the context of carbon capture from the atmosphere and its subsequent use in synthetic fuel production, the authors are indeed referring to CO₂ as the carbon source. Therefore, specifying "carbon dioxide" instead of just "carbon" accurately represents the chemical composition and source of the carbon used in the process.

(5) For technicality, direct air capture (DAC) with sequestration should be direct air carbon capture and storage (DACCS).

(6) One of the key findings in the authors' paper is the effect of constraining or unconstrained CO₂ sequestration.

However, no background to this is given in the introduction. These additions would help readers appreciate the authors' goal of constraining and not constraining CO₂ sequestration. The inclusion of brief discussions in these areas would support and improve what the authors have provided in paragraph 1 on page 5.

Some recent papers on this are as follows, and they could be referred to improve the state-of-the-art.

<https://doi.org/10.1016/j.rser.2023.113578>

<https://doi.org/10.1016/j.ijggc.2022.103766>

Authors could improve their introduction in this regard by:

1. Briefly highlighting some risks and trade-offs associated with different CDRs in this case BECCS and DACCS.
2. How high reliance on CDRs could lead to negative impacts on global energy-land-water system.
3. CDR moral hazard

A similar rationale behind constraining biomass resources should also be briefly added.

(7) "Similarly, when CO₂ sequestration is constrained, total CO₂ sequestration is forced to be less than 3.8 Gt CO₂ per year (in 2050), which is the 10th percentile from the distribution for this variable"

It should be explained how this level of constraint is distributed between fossil CCS, BECCS, and DACCS. Aside from the quantification, authors should indicate whether this distribution is endogenously or exogenously achieved.

Furthermore, some perspective on this level of constraint could be given to inform readers of the extent of CO₂ sequestration in this study compared to previous estimates. For example, the authors did something similar in the case of biomass constraint by indicating how their level is significantly lower compared to previously used estimates. Recent studies have shown CO₂ sequestration of >10 GtCO₂/yr by 2050 for 1.5C with limited or no overshoot. Such studies should be put in perspective against the levels considered by the authors.

(8) “BECCS accounts for all CDR when biomass is unconstrained (UC), whereas DAC accounts for a significant share of CDR when biomass is constrained (BC)”.

Some brief rationale for these observations should follow this sentence for the benefit of the readers.

DACCS can be energy and cost-intensive, and in scenarios where biomass availability is unlimited, it is cheaper to reach net zero by avoiding removals via DACCS to pursue BECCS. However, when biomass availability is highly limited, bioenergy from the electricity and hydrogen sectors reduces, and this increases fossil fuel consumption. As such DACCS deployment increases to offset emissions.

(9) “When CO₂ sequestration (and therefore CDR) is limited (SC), biofuel deployment increases. Although less CDR might reduce bioenergy needed for BECCS, more primary bioenergy is used produce biofuels than would be used by BECCS to offset the associated emissions from an equivalent amount of conventional fuels”.

Some brief rationale for these observations is needed.

Under highly available CO₂ sequestration, the urgency to reduce fossil fuel consumption decreases and this would decrease low or zero-carbon energy including bioenergy. Therefore, when CO₂ sequestration is highly constrained, there is an immediate and significant reduction in fossil fuels which gives room for bioenergy consumption to increase in general to be able to meet climate targets especially when the bioenergy for BECCS reduces in this instance.

(10) “For example, the SBC case requires a ~5x increase in electricity production by 2050 relative to current production, essentially doubling what is needed to realize the assumed ~2.5x increase in electricity final demand due to end use electrification. Even when overall final demand is considerably lower because of demand-side mitigation, electricity production in the SBC case would still expand by ~3.5x relative to today”

These implications can be briefly well supported by putting the results in perspective in terms of cost. What would be the cost implication in such cases where demand for electricity increases 3.5 to 5 times? This provides a stronger argument about 'pressure on the electricity system' as the authors claim.

The cost could be related to what is required for building, operating, consuming fuel, and maintenance.

(11) “Given the biomass and electricity outcomes, the land use requirements in this case are potentially smaller than in the other constrained scenarios”.

It seems the conclusions reached by the authors here are only abstract since their adopted model does not provide results for land use. However, land use is more complicated, and in some cases, there are some counter-intuitive results in land use competition across scenarios where an actual IAM was used.

Authors can use some simple equations to obtain results for land use scarcity for electricity and hydrogen production across the different scenarios and limit their land use effect to just these. If other equations could be obtained to help estimate land use across the scenarios that would be even better. This informs readers of the actual extent to which one scenario reduces land-use trade-offs compared to other scenarios.

Same for water requirements.

Authors could refer to some equations from the paper below

<https://doi.org/10.1038/s41467-023-41107-x>

(12) Fig. 3

1. Could the authors present some of these results in tabular representation in the supplementary file, as these results are key in the discourse?

2. Can some explanations be presented for the significant difference in the share of BECCS elec and BECCS H2 in the UC scenario. As I noticed from the figure, the BECCS H2 is similar in the UC and BC scenario

3. The conventional oil-derived fuel and NG present an interesting result. I am wondering why the production of NG is not more pronounced as against conventional oil (Liquids), despite the presence of unconstrained sequestration to offset emissions of the more carbon-intense fossil fuel. Some comments will be appreciated

(13) Some additional comments:

(1) How robust are the conclusions reached if a sensitivity analysis were to be conducted based on the cost of technologies (and their efficiencies) and the cost of energy feedstocks?

(2) Also, authors would have to double-check and read over the manuscript. In some instances, some prepositions and punctuations are missing, which makes some of those sentences difficult to read and understand at first read. Also, spelling needs to be checked. For example, authors spell unconstrained as unconstrained on page 8, first sentence of last paragraph

(3) If possible, the model, its codes, and all supporting files should be made publicly available online. This ensures transparency, and repeatability, and helps others to learn and come up with alternative pathways in order to push the research field in the right direction.

Reviewer #2 (Remarks to the Author):

The work presents highly scientific value and reveals an interesting approach.

However, there are small aspects that deserve some improvements:

- Abstract- a brief description of the method could be included.
- Graphs - letter of figure 3 and 5 should be bigger (it isn't easy to read)
- Conclusion - The paper could benefit from having a final chapter gathering and resuming the conclusions.

Reviewer #3 (Remarks to the Author):

1. The work is interesting and provides new insights into the options and solutions that different constraints provide in terms of delivering on the remaining demand of liquid fuels and gas while achieving a net-zero emissions energy system. It provides a notable complement to existing literature featuring scenarios leading to a net-zero emissions energy system.

2. Some of the more noteworthy results include the implication that limiting one resource (for example, biomass) may inadvertently put pressure on another (for example, CO₂ sequestration). The study thus demonstrates the inverse relationship between limitations in biomass supply and CO₂ sequestration. Further, the study exhibits the additional pressure on the electricity system when both biomass and CO₂ sequestration are constrained. It is an interesting finding that the SBC case requires a ~5x increase in electricity production by 2050 relative to current production, and that even when overall final demand is considerably lower because of demand-side mitigation, electricity production in the SBC case would still expand by ~3.5x relative to today.

3. The discussion under the heading Future Directions provides a holistic view and provides for a relevant interpretation of the findings as well as the limitations of the findings and how future research can fill the gaps in this regard. This includes connecting detailed process modeling to the types of inputs needed for energy system modeling and the need for higher resolution (including the coupling between the electricity sector and other energy sectors, as well as the implication of energy storage solutions) to provide more granular information for decision-makers while simultaneously informing the parameterization of coarser-resolution models.

4. It might be pertinent to also add a discussion on other types of demand-side measures in addition to those that are mentioned briefly, such as parameters linked to sufficiency which are highlighted as important levers not the least by the IPCC in its AR6 report suite.

5. I further appreciate that the authors comment on how land demand goes beyond bioenergy and electricity expansion (including natural climate solutions and ecosystem services etc.). The authors also highlight that the scenario space considered in net-zero modeling should be enlarged while taking in aspects that are most consequential for other societal priorities.

6. The introduction would be served by providing additional details on the differences in mitigation approaches between different scenarios that provide varying results in terms of remaining liquid fuels and natural gas in the reference studies presented.

7. Section 2 would serve from an analysis of the explanations provided in the reference documents as regard the overall mitigation levels and share of relative mitigation achieved by different fuels/gases.

8. I suggest that Figure 5 be supported by also presenting the amount of remaining fossil-based liquid fuels and natural gas.

9. The discussion around the SBC case would be served by making a comment about how this scenario shifts if carbon-free electricity is not constrained, considering the impact in this scenario associated with producing H₂ for the power sector.

10. The article should include data and make some comments around overall costs of the energy system in the four core cases, including the sensitivity analysis case.

11. Overall, the methodology is sound and for most of the assumptions where references are given, they appear to be based on data from globally recognized institutions.

12. However, considering the importance of technology assumptions and fuel/non-energy costs for the results of the simple model selection choice, there is a considerable lack of references and explanations for the assumptions chosen, both in regard to the technologies as well as the costs.

13. Considering the importance of fuel costs for the results, the choice of static prices as well as the references and rationale for the values set should be included and explained properly. For example, the cost of natural gas appears relatively low at \$6 per GJ, whereas the IMF in its 2023 Update of its Fossil Fuel Subsidies Data (<https://www.imf.org/en/Publications/WP/Issues/2023/08/22/IMF-Fossil-Fuel-Subsidies-Data-2023-Update-537281>) describes natural gas prices of around \$15/GJ in 2030. On the contrary, the biomass prices appear high with \$10 per GJ whereas biomass prices in other studies range from \$2-\$5 /GJ (see e.g. <https://doi.org/10.1016/j.fuproc.2022.107585> ; <https://www.mdpi.com/1996-1073/14/14/4181#>; <https://www.statista.com/statistics/856660/biomass-feedstock-prices-in-the-us-by-product/>)

14. There does not appear to be a clear consistency in regard to whether the assumptions used are taking technological developments over time to 2050 into account. For example, the authors have chosen not to include upstream energy feedstock emissions in its core cases partly due to the assumed implementation of mitigation measures. As an example of the contrary, however, efficiency assumptions for the Conventional, Bio FT, and Bio FT CCS are static on 2020 years level. The authors should be able to

demonstrate consistency between assumptions, or at least acknowledge this deficiency if sufficient relevant data is not deemed available for all modeling inputs.

15. Omitting upstream energy feedstock emissions in the core cases gives a skewed result. While assumed implementation of mitigation measures could be included if properly explained, at least non-CO₂ emissions and emissions associated with land use change should be included in the core cases.

16. The authors have chosen to set a high CO₂ capture rate of 95% for CCS/BECCS without referring to studies that support this level. Most similar studies assume a maximum realistic capture rate of 90%.

Response to Reviewer Comments

Reviewer #1 (Remarks to the Author):

Dear Editor, I have completed my review of the assigned manuscript. Based on the level of availability of CO₂ sequestration and biomass resources, the authors sought to investigate the drivers and implications of alternative routes to emission reduction from significant volumes of hydrocarbon fuels that remain in the energy system during net zero. The authors used a simple structural model of the global energy system based on exogenously specified final energy demands, biomass supply, and CO₂ sequestration. The work is interesting and relevant. However, some reported results would need further discussion and clarification for the benefit of prospective readers. Some minor corrections regarding typos and grammar are required in a few places as well. Please find my comments below, and I look forward to the revised version.

Thank you for the helpful review. We have responded to specific comments and suggestions below.

(1) The context in which 'fuel' in sentence 3 of the abstract is used is ambiguous.

Which types of fuels remain? For example, by the time net zero is reached, coal (which is a 'fuel') would have declined significantly. So, in such scenarios, 'significant coal volumes DO NOT remain even when net zero emissions are reached' compared to the levels today.

Agreed. We have defined “fuels” at first use in the abstract and in the main text. In this paper, “fuels” refers to liquid hydrocarbon fuels and natural gas (whether fossil or non-fossil derived). An objective of the paper is to explain how significant remaining volumes of these fuels, as observed in most scenarios, would be consistent with a net-zero emissions objective.

In the abstract: “Although the use of fuels – specifically liquid fuels and natural gas – declines...”

In the main text: “...there are several approaches to mitigate emissions from remaining fuels – specifically from liquid hydrocarbon fuels and natural gas.”

(2) CO₂ should be defined at first use.

Agreed. We have defined CO₂ at first use in abstract and in main text.

(3) Introduction: paragraph 1.

“At the same time, significant amounts of liquid fuels and natural gas remain in most scenarios at the time of net-zero emissions”

It would help readers if some rationale or reason for this observation in such scenarios at the time of net zero emissions is given after this sentence. One may wonder why significant volumes of hydrocarbon liquid fuels may remain even after net zero. Authors could include reasons such as the energy requirements of hard-to-abate sectors and also high carbon lock-in in some countries/regions/sectors.

Agreed. We have added such a sentence explaining why significant volumes of these fuels remain:

“At the same time, significant amounts of liquid fuels and natural gas remain in most scenarios at the time of net-zero emissions (Luderer, et al., 2018). This result occurs because it may difficult or costly to electrify some “hard-to-abate” sectors, and retaining some fuels while mitigating their emissions may be more cost-effective than other options (Clarke, et al., 2022).”

(4) “Another approach is to combine carbon that is removed from the atmosphere with hydrogen to make a synthetic (non-bio) hydrocarbon fuel (synthetic fuels)”

For clarity, it should be CO₂ with hydrogen instead of carbon with hydrogen. In the context of carbon capture from the atmosphere and its subsequent use in synthetic fuel production, the authors are indeed referring to CO₂ as the carbon source. Therefore, specifying "carbon dioxide" instead of just "carbon" accurately represents the chemical composition and source of the carbon used in the process.

Agreed. We have made this change.

(5) For technicality, direct air capture (DAC) with sequestration should be direct air carbon capture and storage (DACCS).

We have added the acronym “DACCS” at first use. However, it is also useful to retain the acronym “DAC” since DAC is sometimes used to produce CO₂ for utilization rather than for sequestration. Both are used in the paper now.

(6) One of the key findings in the authors' paper is the effect of constraining or unconstrained CO₂ sequestration.

However, no background to this is given in the introduction. These additions would help readers appreciate the authors' goal of constraining and not constraining CO₂ sequestration. The inclusion of brief discussions in these areas would support and improve what the authors have provided in paragraph 1 on page 5.

Some recent papers on this are as follows, and they could be referred to improve the state-of-the-art.

<https://doi.org/10.1016/j.rser.2023.113578>

<https://doi.org/10.1016/j.ijggc.2022.103766>

Authors could improve their introduction in this regard by:

- 1. Briefly highlighting some risks and trade-offs associated with different CDRs in this case BECCS and DACCS.**
- 2. How high reliance on CDRs could lead to negative impacts on global energy-land-water system.**
- 3. CDR moral hazard**

A similar rationale behind constraining biomass resources should also be briefly added.

These issues are taken up more directly in Section 5 on “Societal Tradeoffs, but we agree it would be helpful to mention them briefly upfront as motivation for the scenarios examined. As suggested by the reviewer, constraints on these resources in scenarios likely reflect a range of factors, including concerns about land and water use. In this way, scenarios that vary these resources reflect societal choices about how to use or limit them, as well as other institutional factors that may affect their deployment. We have added text in the introduction and included the references provided.

“The variation in biomass primary energy and CDR, in turn, reflects the uncertainty in how society may decide to use or limit these resources given other factors such as land and water use (Liu, et al., 2023) as well as other institutional factors affecting their deployment (Grant, et al., 2022).”

(7) “Similarly, when CO₂ sequestration is constrained, total CO₂ sequestration is forced to be less than 3.8 Gt CO₂ per year (in 2050), which is the 10th percentile from the distribution for this variable”

It should be explained how this level of constraint is distributed between fossil CCS, BECCS, and DACCS. Aside from the quantification, authors should indicate whether this distribution is endogenously or exogenously achieved.

The constraint is applied to total CO₂ sequestration (Gt CO₂ per year). Therefore, the distribution between fossil CCS, BECCS and DACCS is endogenous. We have clarified this in the text. The resulting distribution can be seen in Figure 3 (see the components of the “captured CO₂” bar).

“In all cases considered here, the sources of captured CO₂ (fossil, biomass, DAC) are determined endogenously by the optimization.”

Furthermore, some perspective on this level of constraint could be given to inform readers of the extent of CO₂ sequestration in this study compared to previous estimates. For example, the authors did something similar in the case of biomass constraint by indicating how their level is significantly lower compared to previously used estimates. Recent studies have shown CO₂ sequestration of >10 GtCO₂/yr by 2050 for 1.5C with limited or no overshoot. Such studies should be put in perspective against the levels considered by the authors.

We agree it would be helpful to provide some perspective on the level of CO₂ sequestration in the scenarios. To develop a benchmark like the one used for biomass (a low end considering real-world factors other than resource availability), we used the “Low Investible Storage Potential” from the Grant et al study cited above. This comparison reveals that the CO₂ sequestration in our constrained case is somewhat lower than the benchmark amount, just as the biomass in our constrained case is somewhat lower than the sustainable biomass benchmark. This comparison confirms that our constrained cases represent reasonable low ends for biomass and CO₂ sequestration for the purposes of this study.

“Similarly, when CO₂ sequestration is constrained, total CO₂ sequestration is forced to be less than 3.8 Gt CO₂ per year (in 2050), which is the 10th percentile from the distribution for this variable. This is lower

than the amount deployed in the “Low Investable Storage Potential” 1.5C case considered by Grant et al (2022).”

(8) “BECCS accounts for all CDR when biomass is unconstrained (UC), whereas DAC accounts for a significant share of CDR when biomass is constrained (BC)”.

Some brief rationale for these observations should follow this sentence for the benefit of the readers.

DACCS can be energy and cost-intensive, and in scenarios where biomass availability is unlimited, it is cheaper to reach net zero by avoiding removals via DACCS to pursue BECCS. However, when biomass availability is highly limited, bioenergy from the electricity and hydrogen sectors reduces, and this increases fossil fuel consumption. As such DACCS deployment increases to offset emissions.

We agree some additional explanation would be helpful. We have added a brief explanation for the result above, consistent with the one suggested by the reviewer (when BECCS is available, it is more cost-effective than DACCS). The second part of the reviewer’s comment – that lowering BECCS increases fossil use, leading to an even greater need to deploy DACCS – is captured later when interactions between the resources are discussed. Specifically, we state:

“Conversely, when biomass supply is limited, use of DAC expands. Since DAC consumes rather than produces energy, it does not displace other fossil energy technologies as BECCS would, thereby increasing the need for CCS directly (on the fossil technologies) or indirectly (by requiring more DAC to offset unabated fossil emissions). In either case, we find that CO₂ sequestration increases as biomass is reduced.”

(9) “When CO₂ sequestration (and therefore CDR) is limited (SC), biofuel deployment increases. Although less CDR might reduce bioenergy needed for BECCS, more primary bioenergy is used produce biofuels than would be used by BECCS to offset the associated emissions from an equivalent amount of conventional fuels”.

Some brief rationale for these observations is needed.

Under highly available CO₂ sequestration, the urgency to reduce fossil fuel consumption decreases and this would decrease low or zero-carbon energy including bioenergy. Therefore, when CO₂ sequestration is highly constrained, there is an immediate and significant reduction in fossil fuels which gives room for bioenergy consumption to increase in general to be able to meet climate targets especially when the bioenergy for BECCS reduces in this instance.

We agree that this result would be helpful to further explain. The following statement, which is meant to illustrate the effect, is included directly after the sentences mentioned by the reviewer:

“For example, if the carbon intensities (per unit energy) of oil and biomass are comparable, then 2 EJ of primary bioenergy are needed to replace 1 EJ of oil-based liquid fuel (assuming a conversion efficiency of ~50% to produce fuel from biomass), while only 1 EJ of primary bioenergy would be needed in the form of BECCS to offset the emissions from 1 EJ of oil-based liquid fuel, assuming all of the CO₂ from

BECCS could be captured and sequestered, which is approximately true for BECCS in power or hydrogen production.”

Regarding the explanation suggested by the reviewer, it is true that constraining CDR makes room for alternative fuels. We have modified the first sentence above (cited by the reviewer) to reflect this. In the second sentence cited by the reviewer, we are calling attention to the fact that, if an equivalent amount of fossil liquids emissions is mitigated by biofuels rather than by relying on BECCS in another sector (as an offset), more biomass primary energy is used when it is used to produce fuels due to the conversion losses associated with fuels production. While the conversion efficiency of biomass for BECCS in power or H₂ production is potentially even lower than that for biofuels production, the CO₂ capture efficiency is higher (most of the carbon in the biomass can be captured), so it is a more efficient use of biomass in the sense that less primary bioenergy is required to produce a unit of emissions mitigation.

(10) “For example, the SBC case requires a ~5x increase in electricity production by 2050 relative to current production, essentially doubling what is needed to realize the assumed ~2.5x increase in electricity final demand due to end use electrification. Even when overall final demand is considerably lower because of demand-side mitigation, electricity production in the SBC case would still expand by ~3.5x relative to today”

These implications can be briefly well supported by putting the results in perspective in terms of cost. What would be the cost implication in such cases where demand for electricity increases 3.5 to 5 times? This provides a stronger argument about 'pressure on the electricity system' as the authors claim.

The cost could be related to what is required for building, operating, consuming fuel, and maintenance.

In general, the cost differences between the scenarios are significant. We have made a choice in this paper not to emphasize cost implications more than other implications in the discussion, recognizing that there are many factors affecting societal choices, some of which (including cost) are discussed in Section 5 on “Societal Tradeoffs”. In addition, cost is typically found to be a less robust outcome across models and the choice of cost metric. Other models may be better suited to evaluating cost or welfare impacts.

That said, we believe the relative differences in cost between the scenarios are meaningful. For this reason, we have reported differences in marginal abatement costs in the SI (Table S12 and Tables S16-20). In response to this comment, we have added text calling attention to differences in cost between the cases in Section 5 and pointed to the marginal abatement costs. We do not have a straightforward way to extract only the electricity costs, but by looking at the electricity prices (same table), we can see that total electricity system cost (if approximated as P*Q) would vary approximately with the scale of electricity production.

(11) “Given the biomass and electricity outcomes, the land use requirements in this case are potentially smaller than in the other constrained scenarios”.

It seems the conclusions reached by the authors here are only abstract since their adopted model does not provide results for land use. However, land use is more complicated, and in some cases, there are some counter-intuitive results in land use competition across scenarios where an actual IAM was used.

Authors can use some simple equations to obtain results for land use scarcity for electricity and hydrogen production across the different scenarios and limit their land use effect to just these. If other equations could be obtained to help estimate land use across the scenarios that would be even better. This informs readers of the actual extent to which one scenario reduces land-use trade-offs compared to other scenarios.

Same for water requirements.

Authors could refer to some equations from the paper below

<https://doi.org/10.1038/s41467-023-41107-x>

Thank you for the reference. We have cited it when discussing the SBC case that relies heavily on electrolytic hydrogen. Regarding the quantitative land use outcomes, it is very difficult to quantify within the current framework, other than to note some of the likely qualitative differences between scenarios. If we exclude a source of land use change in such calculations, or do not properly attribute land use change within the electricity sector to different technologies, which we do not fully characterize in our scenarios, our assessment may not capture the actual differences between scenarios, given potentially significant differences in land use intensities between technologies, as shown for example, in:

<https://journals.plos.org/plosone/article?id=10.1371/journal.pone.0270155>

Regarding water use, that could also depend heavily on rather granular technology choices that we do not represent, such as different cooling technologies in the electric sector. See for example:

Review of Operational Water Consumption and Withdrawal Factors for Electricity Generating Technologies (nrel.gov).

To avoid numerical estimates that might be incomplete or incorrect, we would rather point out the limitations of this study regarding this issue and have done so when land and water requirements are first discussed.

In response to this comment, we also looked at the IPCC AR6 database results to see if there is reported information that would be informative. Unfortunately, only a subset of C1 scenarios reported water withdrawals, and there is not strong correlation with the scenario dimensions we are examining. While most scenarios report land use for bioenergy production, such land requirements scale directly with bioenergy production, as one would expect. To our knowledge, there is no reported information on land use related to other energy sources, so it is not possible to examine all sources of land pressure together from the information available in the database.

(12) Fig. 3

1. Could the authors present some of these results in tabular representation in the supplementary file, as these results are key in the discourse?

Yes, we have added these results in tabular form to the SI (Tables S8-S11).

2. Can some explanations be presented for the significant difference in the share of BECCS elec and BECCS H2 in the UC scenario. As I noticed from the figure, the BECCS H2 is similar in the UC and BC scenario

Yes, we have added the following explanation:

“Since BECCS electricity and BECCS hydrogen provide similar functions in a net-zero energy system (CDR), their relative deployment is determined primarily by cost differences. BECCS hydrogen is slightly more cost-effective under default assumptions, leading it to deploy up until the demand for hydrogen is satisfied, followed by deployment of BECCS electricity. Other assumptions may result in a different cost ordering.”

3. The conventional oil-derived fuel and NG present an interesting result. I am wondering why the production of NG is not more pronounced as against conventional oil (Liquids), despite the presence of unconstrained sequestration to offset emissions of the more carbon-intense fossil fuel. Some comments will be appreciated

The production of natural gas is determined by assumed final demand (exogenous) and any intermediate uses for natural gas (endogenous). The assumed final demand for natural gas in 2050, based on the median values in the IPCC 1.5C (C1) scenarios, is relatively small compared to final liquids demand (30 EJ for natural versus 91 EJ for liquids). The range across IPCC 1.5C (C1) scenarios in 2050 is quite large for both natural gas and liquids (Table S2). We have added a comment on this at the end of Section 3.

“It is also worth noting that the assumed exogenous final demand for natural gas is considerably lower than the final demand for liquid fuels (30 EJ for natural gas versus 91 EJ for liquid fuels), although there is significant variability across IPCC 1.5°C scenarios in 2050 (see Table S2).”

The endogenous intermediate uses of natural gas can be seen in Figure 3. In general, the potential intermediate uses (electricity production, hydrogen production, etc.) are more easily substitutable with carbon-free resources than demands in end use sectors.

(13) Some additional comments:

(1) How robust are the conclusions reached if a sensitivity analysis were to be conducted based on the cost of technologies (and their efficiencies) and the cost of energy feedstocks?

We agree that it would be helpful to comment on the sensitivity of our results to technology assumptions. Toward this end, we have added new sensitivities that vary fuel price assumptions as well as technology cost assumptions, both of which affect technology competition. Specifically, in response to a comment by reviewer #3, we have considered higher natural gas prices and lower bioenergy prices. In addition, based on the reasoning below, we have added a case with higher BECCS electricity and BECCS H₂ costs. In developing a comprehensive response to this question, we have also run other sensitivities and have added new SI figures when the results were noticeably affected by the assumption changes. Across the sensitivity cases we considered, the primary findings of this paper – shown in Figures 4 and 5 – are robust to these assumptions about technology. That said, there are some interesting differences from the core cases, which we have noted where relevant.

In general, while there are many different technology cost assumptions and it is not possible in this paper to consider the sensitivity to each one individually, we can ask which high-level conclusions (such as those in Figure 4) could be sensitive to such assumptions. The UC and SC cases, which rely heavily on bioenergy, are most likely to be sensitive to assumptions about bioenergy. The new fuel price sensitivity considers lower biomass prices, which effectively lowers the cost of deploying bioenergy technologies. The biomass constrained versions of these cases (BC and SBC) effectively consider higher biomass prices (via the shadow price on the biomass constraint). A key finding from this paper (highlighted previously) is that restricting biomass supply (i.e., raising the cost of deploying bioenergy) significantly affects how emissions from remaining fuels are mitigated. Conversely, we find that lowering the biomass price from its default value does not strongly affect the results (see the fuel price sensitivity results in the new Figure S5).

Of course, changes in the cost of specific bioenergy technologies could also affect results, albeit in a more limited way. For example, as noted earlier and now stated in the text, changing the cost of BECCS H₂ versus BECCS electricity could change the cost and resulting deployment ordering of these technologies. In the BECCS sensitivity case (specifically a case with BECCS electricity and BECCS H₂ non-energy costs raised by 50%), we find that deployment of BECCS shifts from electricity and hydrogen to liquids production. Because this affects the sector in which bioenergy preferentially deploys, we have added this case as an explicit technology sensitivity (Figure S6). Despite these interesting differences with the core results, we find that overall results shown in Figure 4 and Figure 5 are robust to these assumptions.

In the BC case, DACCS mitigates a significant share of remaining fuels emissions. The future cost of DACCS is quite uncertain. Since DACCS deployment increases when BECCS is constrained, increasing the cost of DACCS does not fundamentally change the solution, as there are no less expensive available alternatives to mitigate emissions from remaining fuels even with such higher DACCS costs. We have verified this by running a sensitivity with the non-energy cost of DAC increased by 2x. Similarly, lowering the cost of DACCS does not change the solution in the UC case, because DACCS would still typically be more expensive than BECCS. We have verified this by running a sensitivity with the non-energy cost of DAC decreased by 50%. We have not reported these results in separate figures since they are comparable to the relevant core cases but have presented this reasoning as part of the additional material added on technology cost assumptions in the expanded Section S2.

In the SBC case, non-bio synthetic fuels play a more significant role, and the cost of producing such fuels is also quite uncertain. However, in this case, synthetic fuels deploy because other options (BECCS, DACCS, biofuels) are not available. Therefore, raising the cost of producing synthetic fuels increases the system cost at net-zero emissions, but there is no available lower-cost alternative that can be selected given the constraints under which these technologies deploy. We have confirmed this by running a case with arbitrarily high non-energy costs for synthetic fuels production but have not reported results in separate figures since they are comparable to the core cases.

We conclude from this discussion that the main findings of the study are robust to future technology cost uncertainty within reasonable ranges. We have added a version of this discussion to the section in which technology assumptions are described (Section S2).

(2) Also, authors would have to double-check and read over the manuscript. In some instances, some prepositions and punctuations are missing, which makes some of those sentences difficult to read and understand at first read. Also, spelling needs to be checked. For example, authors spell unconstrained as unconstrained on page 8, first sentence of last paragraph

Thank you for mentioning the typo. We have made the correction on p. 8 and have read over the entire revised manuscript to hopefully identify and correct remaining typos.

(3) If possible, the model, its codes, and all supporting files should be made publicly available online. This ensures transparency, and repeatability, and helps others to learn and come up with alternative pathways in order to push the research field in the right direction.

Yes, if the manuscript is accepted, we would upload the model code and relevant data as supplementary files prior to publication.

Reviewer #2 (Remarks to the Author):

The work presents highly scientific value and reveals an interesting approach.

Thank you for the review of this manuscript and for noting its scientific value.

However, there are small aspects that deserve some improvements:

- Abstract- a brief description of the method could be included.

We have now mentioned the illustrative scenario approach. The use of a simple structural model was mentioned previously. I believe the number of words allowed in the abstract may be quite limited. However, if the editor would like us to expand further, we could do so.

- Graphs - letter of figure 3 and 5 should be bigger (it isn't easy to read)

We have enlarged as many of the labels as possible. If the paper is accepted, we could work with the production team to ensure that the figures are readable.

- Conclusion - The paper could benefit from having a final chapter gathering and resuming the conclusions.

Given the space limitations of *Nature Communications*, we have focused the final section on recommended future directions for the research community. However, in response to this comment, we have summarized the scenario analysis at the start of this section.

Reviewer #3 (Remarks to the Author):

1. The work is interesting and provides new insights into the options and solutions that different constraints provide in terms of delivering on the remaining demand of liquid fuels and gas while achieving a net-zero emissions energy system. It provides a notable complement to existing literature featuring scenarios leading to a net-zero emissions energy system.

Thank you for the thoughtful review. We have responded to the specific suggestions and comments below.

2. Some of the more noteworthy results include the implication that limiting one resource (for example, biomass) may inadvertently put pressure on another (for example, CO₂ sequestration). The study thus demonstrates the inverse relationship between limitations in biomass supply and CO₂ sequestration. Further, the study exhibits the additional pressure on the electricity system when both biomass and CO₂ sequestration are constrained. It is an interesting finding that the SBC case requires a ~5x increase in electricity production by 2050 relative to current production, and that even when overall final demand is considerably lower because of demand-side mitigation, electricity production in the SBC case would still expand by ~3.5x relative to today.

We agree that these are some of the more interesting findings that complement existing scenario studies and could inform choices related to the energy transition.

3. The discussion under the heading Future Directions provides a holistic view and provides for a relevant interpretation of the findings as well as the limitations of the findings and how future research can fill the gaps in this regard. This includes connecting detailed process modeling to the types of inputs needed for energy system modeling and the need for higher resolution (including the coupling between the electricity sector and other energy sectors, as well as the implication of energy storage solutions) to provide more granular information for decision-makers while simultaneously informing the parameterization of coarser-resolution models.

Thank you for noting these recommendations for future research community activity. We feel that these are important directions that will enable the modeling community to continue informing important policy and technology decisions as the energy transition continues.

4. It might be pertinent to also add a discussion on other types of demand-side measures in addition to those that are mentioned briefly, such as parameters linked to sufficiency which are highlighted as important levers not the least by the IPCC in its AR6 report suite.

Agreed, we have mentioned sufficiency as one type of demand-side response when discussing the features common to published net-zero scenarios:

“Regardless of the framework used to produce them, several features are common to most net-zero scenarios, such as demand-side changes, including greater deployment of energy efficiency and sufficiency measures, increased electrification of end uses...”

5. I further appreciate that the authors comment on how land demand goes beyond bioenergy and electricity expansion (including natural climate solutions and ecosystem services etc.). The authors also highlight that the scenario space considered in net-zero modeling should be enlarged while taking in aspects that are most consequential for other societal priorities.

Thank you for noting the recommendation to enlarge the scenario space, which we believe is also important for providing decision-makers with a broad view of the different options available and their potential consequences and tradeoffs.

6. The introduction would be served by providing additional details on the differences in mitigation approaches between different scenarios that provide varying results in terms of remaining liquid fuels and natural gas in the reference studies presented.

In general, scenarios with higher final demands for liquids and natural gas would have either less end use electrification and/or greater total final demand. Deployment of hydrogen in end uses is another factor that could affect remaining liquid fuels and natural gas, but it is typically a less important factor than the other two. We have examined the variation in final energy carriers in Table S2, and we have added a brief discussion of these differences in the introduction. However, a more complete analysis of the factors driving differences in the overall amount of remaining fuels is outside the scope of what we can cover in this study.

“Differences in the amounts of remaining liquid fuels and natural gas are apparent in the significant variation in those fuels in the 1.5°C scenarios discussed above. To some extent these changes can be attributed to differences in the extent of electrification and differences in overall final energy demand (Table S2).”

7. Section 2 would serve from an analysis of the explanations provided in the reference documents as regard the overall mitigation levels and share of relative mitigation achieved by different fuels/gases.

Agreed, we have added additional explanation where the drivers are clearest from the underlying studies, particularly for the studies in which this author team has participated. However, looking across studies, it would be difficult to say that all the drivers are well understood or easily extracted from the underlying reference documents. In that sense, the lack of a complete explanation serves as motivation for this study, which is why we have included this section prior to the development of the illustrative scenarios. We conclude the paragraph discussing these studies by saying that “what is currently lacking is a unified explanation for these differences in terms of underlying drivers.” The framework that we propose in the paper is meant to help build an understanding of some of the key drivers that may be useful to those developing or interpreting scenarios.

8. I suggest that Figure 5 be supported by also presenting the amount of remaining fossil-based liquid fuels and natural gas.

We agree that total liquids and total natural gas, as well as fossil liquids and fossil natural gas, could be helpful when interpreting this chart. To avoid cluttering the figure with multiple additional metrics, we have placed this information in a new table (Table S15) and referred to this in the caption of Figure 5.

9. The discussion around the SBC case would be served by making a comment about how this scenario shifts if carbon-free electricity is not constrained, considering the impact in this scenario associated with producing H₂ for the power sector.

These differences can be seen by comparing Figures S1 and S4. Overall, the results of the SBC case are similar regardless of whether carbon-free electricity (CFE) is constrained, but there is less hydrogen and electricity in the unconstrained CFE version since these sectors are less directly coupled, as suggested by the reviewer. We have now mentioned this where the SBC case is discussed around Figure 5 in the main text.

“When biomass supply and CO₂ sequestration are both forced to be limited (as in the SBC case), the production of synthetic fuels requires a significant increase in hydrogen production. Increased hydrogen production, in turn, leads to an increase in electricity generation, effectively defining a reciprocal relationship between the use of these resources and the size of the electricity system. When carbon-free electricity is unconstrained, the growth in electricity and hydrogen use is less extreme as there is less direct coupling of these sectors, but the general result holds (compare Figures S1 and S4).”

10. The article should include data and make some comments around overall costs of the energy system in the four core cases, including the sensitivity analysis case.

We agree that some discussion of cost is warranted. However, as noted in response to reviewer #1, we have made a choice in this paper not to focus more heavily on cost differences than other differences, recognizing that there are many factors affecting societal choices, and that cost outcomes may be less robust than other outcomes across modeling approaches and alternative cost metrics. Ultimately, other approaches may be better suited to assess cost or welfare outcomes.

That said, we believe the relative differences in cost between the cases are meaningful. Marginal abatement costs are reported in Table S12, and we have now added this information for the sensitivity cases as well (Tables S16-S20). In addition, we have now mentioned the differences in these cost outcomes in Section 4 and have also incorporated these differences into the discussion in Section 5 on “Societal Tradeoffs”.

11. Overall, the methodology is sound and for most of the assumptions where references are given, they appear to be based on data from globally recognized institutions.

Thank you for reviewing the assumptions. We have tried to use widely cited sources and have documented the key assumptions and sources in the SI. We have also tried to maintain consistency by relying on the same source within a given sector, and across sectors where this is feasible. For example, due to its global focus and the breadth of reporting across sectors (albeit sometimes in different reports), we have used IEA information for each sector, either as a main or supporting source.

Furthermore, in response to later comments, we have tried to validate the main source for each sector with other recent and reputable sources that would be independent of the main source. It should be noted that the amount of available information and the quality of this information varies by technology and sector, given that there is significantly less experience with some technologies, such as many of the advanced low-emission fuel production technologies.

12. However, considering the importance of technology assumptions and fuel/non-energy costs for the results of the simple model selection choice, there is a considerable lack of references and explanations for the assumptions chosen, both in regard to the technologies as well as the costs.

Regarding the selection of technologies, we have attempted to represent major technology classes that would be most relevant for understanding the overall transition choices discussed in this paper. However, in keeping with the overall approach (development of a simple and transparent structural model), we have refrained from including multiple process configurations within a broader technology class (e.g., multiple 2G biofuel production processes). We have added discussion of this general approach to technology characterization in Section S2.

Regarding references, the primary sources used to develop the assumptions are provided in Section S2. In response to this comment, we have added other references that independently support many of the main assumptions used in each sector. These sources include publications from globally recognized institutions such as IEA, NREL, and NETL. Specifically, the following sources have been added to the main sources:

Supporting source for electricity production technologies:

<https://www.iea.org/data-and-statistics/data-product/global-energy-and-climate-model-2023-key-input-data>

Supporting source for hydrogen production technologies:

<https://pubs.acs.org/doi/10.1021/acs.est.3c03751>

The source above extends the NREL H2A assumptions to 2050:

<https://www.nrel.gov/hydrogen/h2a-production-models.html>

Supporting source for hydrogen production from NG:

https://netl.doe.gov/projects/files/ComparisonofCommercialStateofArtFossilBasedHydrogenProductionTechnologies_041222.pdf

Supporting sources for biofuels production:

https://www.ieabioenergy.com/wp-content/uploads/2020/02/T41_CostReductionBiofuels-11_02_19-final.pdf

Techno-economic prospects for producing Fischer-Tropsch jet fuel and electricity from lignite and woody biomass with CO2 capture for EOR - ScienceDirect

Supporting Sources for DAC:

<https://www.netl.doe.gov/energy-analysis/details?id=d5860604-fbc7-44bb-a756-76db47d8b85a>

<https://netl.doe.gov/energy-analysis/details?id=36385f18-3eaa-4f96-9983-6e2b607f6987>

Although our assumptions are supported by multiple reputable sources, all assumptions should still be considered illustrative given regional, temporal and process heterogeneity that cannot be represented in this simple framework. We have stated this caveat prior to summarizing the cost assumptions.

13. Considering the importance of fuel costs for the results, the choice of static prices as well as the references and rationale for the values set should be included and explained properly. For example, the cost of natural gas appears relatively low at \$6 per GJ, whereas the IMF in its 2023 Update of its Fossil Fuel Subsidies Data (<https://www.imf.org/en/Publications/WP/Issues/2023/08/22/IMF-Fossil-Fuel-Subsidies-Data-2023-Update-537281>) describes natural gas prices of around \$15/GJ in 2030. On the contrary, the biomass prices appear high with \$10 per GJ whereas biomass prices in other studies range from \$2-\$5 /GJ (see e.g. <https://doi.org/10.1016/j.fuproc.2022.107585> ; <https://www.mdpi.com/1996-1073/14/14/4181#>; <https://www.statista.com/statistics/856660/biomass-feedstock-prices-in-the-us-by-product/>)

Thank you for calling attention to the issue of fuel prices. Because fuel prices will vary significantly by region and over time, it makes sense to consider the sensitivity to prices, and we have added a new sensitivity case with higher prices for natural gas and lower prices for bioenergy, consistent with the suggestion above. We find that the main conclusions of this study are robust to fuel price assumptions, but there are some interesting differences from the core cases that we now highlight. In addition, we have added a substantial discussion of fuel price assumptions (based on the discussion below), including additional references, in Section S2.

In general, when selecting assumptions for the default fuel prices for this study, we need to consider that we are exclusively considering net-zero scenarios in which natural gas demand, and therefore prices are likely to be lower than today, and conversely that biomass demand, and therefore prices are likely to be higher than today, all else equal. As an example, the IEA Net-Zero study that we cite elsewhere in the paper states that “The rapid drop in oil and natural gas demand in the NZE [Net Zero Emissions scenario] means that... [p]rices are increasingly set by the operating costs of the marginal project required to meet demand, and this results in significantly lower fossil fuel prices than in recent years.” Table 2.1 in the cited IEA study shows prices in 2050 converging across regions at values well below \$6 per MMBtu, which is approximately our default assumption. For this reason, we believe our default assumption is reasonable, and we have added some additional explanation along these lines. That said, as a general matter, it is appropriate to consider a case in which the spread between gas and oil prices is smaller,

because this could change whether fossil liquids or natural gas emissions are mitigated first. This can be accomplished by raising the fossil NG price and we have used the reviewer's suggestion of \$15 per GJ for this sensitivity, noting that this is consistent with other sources (e.g., IMF).

Turning to biomass prices, some of the sources provided by the reviewer seem to primarily rely on waste streams, such as agriculture and forest residues. These are important biomass feedstock sources; however, their supplies are limited globally and regionally. Given our interest in studying future scenarios with significantly greater bioenergy deployment, we want to consider biomass supplies beyond residues, at which point the biomass prices may increasingly reflect the production cost of dedicated energy crops and logs. EMF-33 is one of the more recent multi-model efforts that has examined biomass supply in mitigation scenarios. Rose et al 2022

(<https://link.springer.com/article/10.1007/s10584-022-03336-9>) shows global supply curves for bioenergy up to several hundred EJ in 2050. The study evaluated biomass supplies at prices ranging from \$3 to \$15 per GJ in 2005 dollars (approximately \$5 to \$23 per GJ in current dollars), depending on the model and the amount of biomass supplied. These supply curves suggest that 150-200 EJ/year is available at prices ranging from \$3 to \$8 per GJ in 2005 dollars (approximately \$5 to \$12 per GJ in current dollars). As another point of comparison, the EPRI results in Figure 1 (Blanford et al, 2022) estimated biomass market prices from approximately \$9 to \$29/GJ for 2050 across its three U.S. net zero by 2050 scenarios. Considering this information, we believe the default assumption of \$10 per GJ is reasonable, and we have added some additional explanation for this choice along the lines above to Section S2. That said, a lower biomass prices sensitivity is appropriate, and we have used \$5 per GJ for the sensitivity case, consistent with the reviewer's suggestion and the low end from EMF-33.

We have incorporated these price assumptions into an alternative fuel price sensitivity, shown in Figure S5. Interestingly, in this case, we do see substitution occurring between fossil natural gas and alternative gas technologies (in SC and SBC), with more oil remaining than in the corresponding core cases. We have noted this in the main text where we discuss the preference for displacing fossil liquids over fossil natural gas in the core cases. With lower biomass prices, we also find more deployment of primary bioenergy (relative to the corresponding core case) in the case in which biomass deployment was found to be highest (SC). Despite these interesting differences, the overall decomposition of remaining fuels mitigation (Figure 5) is largely unchanged from the core cases.

14. There does not appear to be a clear consistency in regard to whether the assumptions used are taking technological developments over time to 2050 into account. For example, the authors have chosen not to include upstream energy feedstock emissions in its core cases partly due to the assumed implementation of mitigation measures. As an example of the contrary, however, efficiency assumptions for the Conventional, Bio FT, and Bio FT CCS are static on 2020 years level. The authors should be able to demonstrate consistency between assumptions, or at least acknowledge this deficiency if sufficient relevant data is not deemed available for all modeling inputs.

We agree that consistency regarding technological evolution is desirable. In general, we have sought to use cost and performance information for the year 2050 or comparable "long-term" or "Nth-of-kind" (NOAK) cost estimates to consistently incorporate evolution in technology to the extent possible given the information available in the underlying sources. One of the rationales for selecting the same source

for all technologies in each sector (and across sectors where possible) is that the approach to technology evolution would be most consistent within a given source. However, this is not always possible, and even where it is, we would like to draw upon other sources to avoid overreliance on any single source.

In general, we do not expect cost reductions (or efficiency improvements) to be the same across technologies, even if the approach for incorporating technology evolution is comparable. The extent of technology evolution will depend on several factors including scalability and the level of current maturity of the technology or its components. While a complete assessment of technology evolution is beyond the scope of what we can do in this paper, we have tried to adopt consistent assumptions regarding technology evolution through the choice of sources and by harmonizing the year or type of estimate (e.g., NOAK) when looking across sources. We have clarified our approach for handling technology advancement in the section in which model assumptions are discussed (Section S2).

15. Omitting upstream energy feedstock emissions in the core cases gives a skewed result. While assumed implementation of mitigation measures could be included if properly explained, at least non-CO₂ emissions and emissions associated with land use change should be included in the core cases.

There are number of issues at play here, which we can unpack. However, at the highest level, it should be kept in mind that we have considered both cases – a case with no upstream emissions and a case with relatively high upstream emissions – and the main findings of the study do not depend on this choice. This can be seen by comparing Figure S1 and Figure S3, or by looking across these two cases in Figure 5. Overall, the case with upstream emissions factors is more stringent (effectively it is a net-zero GHG rather than net-zero CO₂ case). In addition, when there are emissions associated with bioenergy, there is slightly less bioenergy in the SC case than in the corresponding core case (SC is the case that deploys the most bioenergy). In response to this comment, we have clarified our approach in the main text and have significantly expanded the discussion of these issues and included additional references in the section in which model assumptions are discussed (Section S2). We have also noted in the main text that the primary findings of this study are robust to the choice of emissions factors and pointed the reader to the relevant figures.

Turning to the specific issues, starting with bioenergy, land use change emissions related to bioenergy production are appropriate to include but are assumed to be zero as a default value in our core cases, with positive values considered in a sensitivity case. A central estimate near zero is appropriate when a large share of the biomass supply is assumed to come from waste streams and second-generation (2G) feedstocks, which is generally observed in strong mitigation scenarios (Rose, et al., 2022; Blanford, et al., 2022). Results for 2G biofuels can be contrasted with results for first-generation crops, such as corn used for ethanol, which show more consistently positive land use change emissions coefficients. We previously cited Field et al (first link below) to explain our default assumption, but we have added several other references to support this choice:

<https://www.pnas.org/doi/epdf/10.1073/pnas.1920877117>

<https://www.worldscientific.com/doi/10.1142/S2010007822500087>

<https://pubs.acs.org/doi/full/10.1021/es5052588>

Soil carbon sequestration and land use change associated with biofuel production: empirical evidence - Qin - 2016 - GCB Bioenergy - Wiley Online Library

It should be noted that while we have considered zero and positive land use change coefficients, negative land use change emissions are also possible and imply net carbon sequestration on land without the use of CCS (BECCS). Negative land use change emissions (net carbon sequestration on land) can occur when the land use and management changes associated with increasing biomass production result in increases in below and above ground carbon stocks that more than offset the decreases in carbon stocks from any land displaced directly or indirectly by increased biomass production. The variability in these estimates around zero for second-generation biomass crops (Field, et al., 2020; Mignone, et al., 2022; Dwivedi, et al., 2015; Qin, et al., 2016) is what led us to select zero as a default assumption in our core cases, but as noted, we examine a sensitivity case with a positive assumed emission factor for biomass production.

Turning to non-CO₂ emissions, while CO₂ emissions across the life cycle are relevant to specifying emissions in a net-zero CO₂ system, the inclusion of non-CO₂ emissions is effectively a question about scenario design. In the IPCC 1.5C scenarios, while energy-related CO₂ emissions are typically near zero in 2050 (Table S1), non-CO₂ forcing is still positive when net-CO₂ emissions are attained, meaning that these scenarios do not attain net-zero GHG emissions until well after 2050 (Riahi, et al., 2022). Including all non-CO₂ emissions related to the energy system would shift the focus from a net-zero CO₂ system (roughly aligned with the IPCC 1.5C scenarios in 2050) toward a net-zero GHG system, which would be more stringent. For these reasons, we do not include upstream emissions factors from fossil fuels in the core cases but instead include them in a sensitivity case. As a separate matter, it should be noted that energy-related non-CO₂ emissions factors would likely be considerably lower than today due to changes in production driven by mitigation incentives in the type of stringent cases we are considering here. In this sense, our choice of emissions factors in the sensitivity case could be considered to be on the high end. Thus, taken together, the two sets of assumptions about upstream emissions factors examine reasonable variation in the definition of a net-zero system, spanning approximately net-zero CO₂ and net-zero GHG energy systems. The main findings of this study are robust to choices about these upstream emissions factors (compare Figures S1 and S3, and the two cases within Figure 5).

We have explained and clarified our choices along the lines above in the significantly expanded model assumptions section (Section S2) and have rewritten the relevant explanation in the main text (Section 2).

16. The authors have chosen to set a high CO₂ capture rate of 95% for CCS/BECCS without referring to studies that support this level. Most similar studies assume a maximum realistic capture rate of 90%.

It is correct that prior studies have often assumed 90% capture rates, but this is not a technical limit, and this assumption is less defensible in more stringent scenarios. That is, the strong incentives to reduce remaining emissions in net-zero scenarios are likely to make higher capture rates more cost-effective. In general, the capture rate could be viewed as endogenous to the scenario. While it is not possible in our study to optimize for the capture rate, several studies have assumed or assessed the possibility of

capture rates significantly higher than 90%, in some cases approaching 100% capture (sometimes referred to as “Deep CCS”). For example, see:

Deep CCS: Moving Beyond 90% Carbon Dioxide Capture | Environmental Science & Technology (acs.org)

<https://www.sciencedirect.com/science/article/pii/S1750583620306642>

<https://www.sciencedirect.com/science/article/pii/S1750583621002255>

The Brandl et al study cited above specifically states “in no case, was a capture rate of 90% found to be optimal, with capture rates of up to 98% possible at a relatively low marginal cost,” and the Du et al study states that “power plants can achieve zero-emissions [effectively 100% capture] with CCS at competitive costs”. In our study, we have chosen 95% to be in between the more historically common but conservative 90% assumption and the more scenario-consistent but ambitious deep CCS assumptions. It could be noted as well that NETL assumes 95% capture in its recent “baseline study” on CCS (NETL Baseline Study Updated to Include the Performance and Cost of High Carbon Capture Rates for Power Generation Systems | netl.doe.gov).

In response to this comment, we have added a summary of our rationale for the choice of 95% capture rate (along the lines above) to the section in which the CCS capture rate assumption is stated (Section S2).

REVIEWERS' COMMENTS

Reviewer #1 (Remarks to the Author):

I really appreciate the authors' careful attention to all my comments. I am satisfied with revisions made, and I RECOMMEND THE PAPER FOR PUBLICATION in its current form.

Reviewer #2 (Remarks to the Author):

All the questions were adequately answered.

Reviewer #3 (Remarks to the Author):

The authors have acted upon and answered all my previous comments thoughtfully and thoroughly. I have no further comments and I recommend the editor to accept the manuscript for publication